# Advances in 2D Group IV Monochalcogenides: Synthesis, Properties, and Applications

**DOI:** 10.3390/ma18071530

**Published:** 2025-03-28

**Authors:** Angel-Theodor Buruiana, Claudia Mihai, Victor Kuncser, Alin Velea

**Affiliations:** 1National Institute of Materials Physics, Atomistilor 405A, 077125 Magurele, Romania; angel.buruiana@infim.ro (A.-T.B.); claudia.mihai@infim.ro (C.M.); vkuncser@infim.ro (V.K.); 2Faculty of Physics, University of Bucharest, Atomistilor 405, 077125 Magurele, Romania

**Keywords:** 2D materials, group IV monochalcogenides, GeS, GeSe, GeTe, SnS, SnSe, SnTe, anisotropic properties

## Abstract

The field of newly developed two-dimensional (2D) materials with low symmetry and structural in-plane anisotropic properties has grown rapidly in recent years. The phosphorene analog of group IV monochalcogenides is a prominent subset of this group that has attracted a lot of attention because of its unique in-plane anisotropic electronic and optical properties, crystalline symmetries, abundance in the earth’s crust, and environmental friendliness. This article presents a review of the latest research advancements concerning 2D group IV monochalcogenides. It begins with an exploration of the crystal structures of these materials, alongside their optical and electronic properties. The review continues by discussing the various techniques employed for the synthesis of layered group IV monochalcogenides, including both bottom-up methods such as vapor-phase deposition and top-down techniques like mechanical and/or liquid-phase exfoliation. In the final part, the article emphasizes the application of 2D group IV monochalcogenides, particularly in the fields of photocatalysis, photodetectors, nonlinear optics, sensors, batteries, and photovoltaic cells.

## 1. Introduction

The development of two-dimensional (2D) materials has led to significant advancements in various technological fields due to their unique electronic, optical, and mechanical properties. Among them, graphene, transition metal dichalcogenides (TMDs) such as MoS_2_, hexagonal boron nitride (h-BN), MXenes, and phosphorene have attracted considerable interest for their potential in nanoelectronics, energy storage, sensors, and optoelectronic applications [1,2,3,4].

The synthesis of these materials can generally be categorized into top-down and bottom-up approaches. Top-down exfoliation methods involve the delamination of bulk layered materials into thinner sheets and include mechanical exfoliation, liquid-phase exfoliation, electrochemical exfoliation, and bipolar exfoliation. Mechanical exfoliation, commonly performed using the Scotch tape method, yields high-quality monolayers but is limited by scalability. Liquid-phase exfoliation, utilizing ultrasonication in solvents, enables large-scale production but often results in reduced flake size and layer control. Electrochemical exfoliation, which involves intercalation of ions followed by exfoliation under an applied potential, allows better control over layer thickness. Bipolar exfoliation, a relatively recent approach, combines anodic and cathodic reactions to achieve efficient exfoliation with minimal defects. In contrast, bottom-up synthesis methods such as chemical vapor deposition (CVD), pulsed laser deposition (PLD), and magnetron sputtering allow for precise control over thickness and crystallinity but often require high-temperature conditions and complex processing steps [2,5,6,7,8,9,10].

Within this context, group IV monochalcogenides (e.g., GeS, GeSe, SnS, and SnSe) have emerged as materials of interest due to their unique orthorhombic structures, anisotropic properties, and eco-friendly nature. These compounds, with puckered layers resembling black phosphorus, also exhibit abundance in the Earth’s crust and cost-effectiveness, making them attractive for a range of sustainable applications [5,11,12,13,14,15,16].

Further interest has been driven by the diversity of crystalline phases observed in certain compounds, such as GeTe and SnTe, which transition between rhombohedral, orthorhombic, and cubic structures under varying conditions. These transformations directly influence material properties, highlighting the versatility of group IV monochalcogenides (group IV MCs) [17,18].

In addition to their structural diversity, these materials have demonstrated exceptional functional properties. Bulk SnSe, for example, holds the record for the highest thermoelectric figure of merit (ZT) of 3.1 at ~780 K, surpassing conventional thermoelectric materials such as Bi_2_Te_3_ [19] and other emerging thermoelectric compounds such as SnTe and Te-based solid solutions [20,21]. Additionally, 2D SnSe monolayers have shown promise in photodetectors, photovoltaics, and thermoelectric applications [22,23,24], while SnSe nanoflakes have exhibited memristive behavior at room temperature, with a threshold voltage of 3 V and an operating current of 10^−4^ A, indicating potential for neuromorphic computing [13].

Similarly, SnS has been explored for tunable bandgap photodetectors, demonstrating long-term stability in acidic and neutral electrolytes [25]. Its efficient ion intercalation enables energy storage applications, with large interlayer spacing facilitating Na^+^ and Li^+^ trapping. Moreover, its volume expansion of ~250% makes it more suitable for repeated cycling compared with its SnS_2_ allotrope [26].

SnTe is a narrow-bandgap semiconductor (0.18 eV in bulk), making it a strong candidate for mid-infrared detection [27]. Its high dielectric constant (ε = 45), multiple symmetry surfaces, and small electron-hole effective masses contribute to its unique electronic properties. In its 2D form, SnTe is a topological crystalline insulator with mirror symmetry, characterized by a mirror Chern number |ηM| = 2, resulting in an even number of Dirac cones in the Brillouin zone [28].

Additionally, monolayers of GeS, GeSe, SnS, and SnSe exhibit substantial piezoelectric properties, making them attractive for electromechanical applications [16,29]. Moreover, these materials play an important role in energy storage systems, with applications spanning from supercapacitors to battery electrodes [30,31,32].

Defect engineering plays an important role in tailoring the electronic, optical, and catalytic properties of 2D group IV monochalcogenides, significantly expanding their potential applications [33]. Atomic vacancies, heteroatom substitutions, and other structural defects can serve as active sites for catalysis, influence charge transport, and enhance ion storage capabilities. These defects introduce localized midgap states that modulate electronic band structures, affecting electrical conductivity and optoelectronic behavior. Furthermore, controlled defect introduction has been exploited to improve thermoelectric performance by reducing lattice thermal conductivity. Similar strategies have been extensively studied in transition metal dichalcogenides, where defect engineering has demonstrated the ability to fine-tune material properties for specific technological applications, such as sensors, memristors, and energy storage devices [34].

Despite these promising properties, several challenges must be addressed to fully exploit their potential. The scalability and stability of these materials remain important issues as oxidation and degradation under ambient conditions can limit their long-term applicability. The synthesis of group IV monochalcogenides presents challenges due to their high exfoliation energy. Approaches such as chemical vapor deposition, solution-phase techniques, and mechanical exfoliation have been employed to overcome these limitations. However, they require further optimization to achieve large-area, high-quality films with controlled layer thickness.

This review provides a comprehensive overview of recent advancements in group IV monochalcogenides, focusing on their structural and electronic properties. It explores the main synthesis techniques, including their advantages and limitations, and discusses the characterization methods for understanding anisotropic properties. Furthermore, the review explores emerging applications, ranging from optoelectronics and spintronics to energy storage and nonlinear optics, emphasizing the potential impact of these materials on next-generation technologies.

## 2. Synthesis Methods

The synthesis technique used to obtain group IV monochalcogenides is a fundamental aspect of device development as it significantly influences the material’s optoelectronic properties and, consequently, the performance of the final device. This section systematically examines bottom-up and top-down synthesis techniques, focusing on chemical vapor deposition for bottom-up fabrication and liquid-phase exfoliation (LPE) alongside mechanical exfoliation (ME) for top-down approaches. Each method’s principles, advantages, limitations, and material-specific applications are analyzed in detail, with comparative insights into their suitability for producing high-quality monolayers and thin films.

### 2.1. Bottom-Up Techniques Through Vapor-Phase Deposition

CVD offers a bottom-up approach for synthesizing large-area, high-quality 2D group IV monochalcogenides with controlled thickness and crystallinity. This technique typically involves the vaporization of solid precursors or the chemical reaction of gaseous species at high temperatures, followed by condensation onto a target substrate. Key parameters influencing CVD growth include the precursor composition, substrate type, temperature, pressure, and carrier gas flow rates. Unlike LPE and mechanical exfoliation, CVD enables precise control over thickness and morphology, facilitating the direct integration of 2D group IV monochalcogenides into electronic and optoelectronic devices. However, challenges such as optimizing growth conditions for uniform monolayer coverage and preventing contamination must be addressed to achieve scalable and reproducible synthesis of these materials. A typical synthesis process is shown in Figure 1. The synthesis begins with the careful selection and placement of precursors within a quartz tube reactor. These precursors, often in the form of elemental powders or halide compounds, are positioned inside the furnace to ensure optimal vaporization. Depending on the specific material being synthesized, direct sublimation of bulk powders, such as GeS or SnSe, may be employed, or precursor reactions, such as the conversion of GeI_2_ with selenium vapor, can be utilized to generate the desired gaseous species. Once the precursors are in place, an inert or reactive carrier gas, typically argon, nitrogen, or hydrogen, is introduced into the system. This controlled gas flow plays an important role in transporting the vaporized species toward the deposition zone while maintaining the stability of the reaction environment. As the furnace temperature gradually increases, the precursors undergo sublimation or thermal decomposition, releasing vapor-phase molecules into the reactor. The precise control of temperature, often ranging between 400 °C and 700 °C, ensures efficient precursor activation while preventing unwanted side reactions that could lead to impurity formation. As the precursor vapors reach the cooler deposition region, they begin to condense onto the selected substrate, which can range from sapphire and SiO_2_/Si to mica or graphite. This nucleation process marks the onset of 2D layer formation, with the choice of substrate significantly influencing the crystal orientation and morphology of the resulting nanosheets. By carefully adjusting gas flow rates, pressure, and deposition time, we can fine-tune the growth process to achieve monolayers or few-layer nanosheets with large lateral dimensions and high crystallinity. Once the synthesis is complete, the system is gradually cooled to room temperature under an inert atmosphere to preserve the structural integrity.

The following sections provide a detailed overview of recent advancements in the CVD synthesis of GeS, GeSe, GeTe, SnS, SnSe, and SnTe, highlighting the parameters that govern their growth and the quality of the resulting materials.

#### 2.1.1. GeS

GeS has been synthesized in 2D form via vapor-phase deposition using GeS powder as the source. GeS powder is heated to sublimation in a furnace, and an inert carrier gas transports the vapor to a cooler region where it condenses into layered GeS sheets. This approach produces flower-like assemblies of single-crystalline GeS “nanoflowers”, composed of nanosheet petals ~20–30 nm thick and extending up to ~100 µm laterally. Growth at reduced pressure (diffusion-limited regime) is important for high-quality sheets; the GeS nanosheets grow rapidly (~3–5 µm/min) yet maintain good crystallinity when mass transport is the rate-limiting step. The obtained GeS sheets are highly crystalline and anisotropic, with strong optical absorption, making them promising for photovoltaics [36].

#### 2.1.2. GeSe

Large-area GeSe nanosheets have been grown by atmospheric pressure CVD using halide precursors. One study employed germanium(II) iodide (GeI_2_) and elemental selenium as precursors: GeI_2_ (sublimation ~240 °C) is placed at ~500 °C in a tube furnace while Se (melting ~220 °C) is heated to 410–460 °C upstream, with Ar/H_2_ carrier gas (3/7 sccm) at ambient pressure. Various substrates (Ge(100), GaAs, c-cut sapphire, and HOPG) were used to nucleate GeSe; a typical growth at ~420 °C on Ge yielded ultrathin rectangular GeSe crystals tens of microns across. The as-grown GeSe is highly crystalline α-phase, as evidenced by its sharp Raman peaks (FWHM ~4.8 cm^−1^, even narrower than bulk crystal) and stoichiometric EDS signal. The sheets are atomically layered and anisotropic, forming either flat platelet morphologies or lamellar flower-like structures depending on the substrate, with low defect densities and confirmed composition [37]. Furthermore, Liu et al. used high purity Ge and Se powders to obtain GeSe nanoplates with lateral dimensions on the order of tens of micrometers and thicknesses ranging from a few nanometers up to approximately 30 nm, depending on the specific growth parameters [38].

#### 2.1.3. GeTe

Two-dimensional GeTe has been successfully synthesized by atmospheric CVD as ferroelectric nanoplates. For example, APCVD growth on freshly cleaved mica substrates yields single-crystalline α-GeTe nanosheets up to ~30 µm in lateral size and as thin as ~8–9 nm. In this method, powdered Ge and Te sources are vaporized in a one-zone furnace at ambient pressure (with the substrate downstream), and careful substrate pre-annealing is used to promote lateral growth while reducing nucleation density. The resulting GeTe sheets have a rhombohedral (α) structure known for room-temperature ferroelectricity, and piezoresponse measurements confirm ferroelectric domains in the as-grown layers. GeTe nanosheets exhibit good crystallinity and uniformity, providing an accessible route to integrate 2D GeTe in ferroelectric devices [39,40].

#### 2.1.4. SnS

Scalable growth of SnS monolayers and few-layer nanosheets has been achieved at ultralow temperatures by CVD. A recent study reported centimeter-scale 2D SnS films grown at just 200 °C, which enabled direct deposition on polymer substrates (e.g., polyimide). The process uses a two-step approach: first, an e-beam evaporated Sn thin film (approximately a few nm) is deposited on the target substrate, then sulfur vapor (from S powder) reacts with it in a CVD furnace. Growth occurs under low pressure (~110 mTorr) with an Ar flow (~100 sccm), and the furnace is ramped to ~250 °C for tens of minutes to convert the Sn film to SnS. This yields uniform SnS layers over >1.5 cm^2^ areas, with thicknesses on the order of 10–30 nm (down to a few layers in thinner regions) and excellent layer continuity. The SnS films are highly oriented (zigzag axis out of plane) and single phase, exhibiting the expected piezoelectric response (confirmed by PFM maps) and mechanical flexibility (maintaining integrity under kirigami cutting). Such SnS nanosheets show potential for wearable piezoelectric devices, combining large area and good crystalline quality [41].

#### 2.1.5. SnSe

Several CVD routes have been developed to grow high-quality SnSe, an orthorhombic layered semiconductor [42]. Using a low-pressure vapor transport method, Chiu et al. grew 2D SnSe crystals with lateral sizes up to ~23 µm and thickness down to ~2 nm (three to four layers) [43]. In this approach, Sn and Se precursors (e.g., SnSe powder or elemental Sn and Se) are vaporized in a tube furnace under reduced pressure, and parameters like substrate surface treatment, growth temperature, and pressure are tuned to favor thin-film formation. The synthesized SnSe flakes are single crystalline and exhibit in-plane ferroelectric polarization domains observable by PFM, indicating excellent crystalline order even at a few-layer thickness. In another paper, a solution-based single-source precursor method enabled wafer-scale (6-inch) SnSe films via CVD-like thermal decomposition, yielding continuous polycrystalline SnSe with uniform optical/electrical properties. These SnSe nanosheets are highly promising, with demonstrated broadband photodetector arrays and stable performance in air due to their high quality and controlled growth process [44]. For SnSe, Zhao et al. [45] synthesized nanoflakes on mica substrates, achieving lateral sizes of 1 to 6 microns with a thickness of just 6 nm using Physical Vapor Deposition (PVD). Another study by Pei et al. [46] demonstrated that monocrystalline SnSe nanoplates, epitaxially grown on molten PDMS, exhibited lateral dimensions of 5 to 15 microns and thicknesses ranging from 9 to 20 nm.

#### 2.1.6. SnTe

Although SnTe is a non-van der Waals (rock salt) crystal in bulk, 2D SnTe nanosheets have been grown by APCVD through careful control of orientation. Su et al. report the atmospheric CVD growth of single-crystalline SnTe plates on inert substrates, using different tin precursors to direct the formation of either (100)-oriented square nanosheets or (111)-faceted nanosheets [47]. In their setup, a Sn-containing precursor (such as SnI_4_ or SnCl_2_) and Te powder are reacted at elevated temperature (e.g., ~600–700 °C) in the presence of H_2_/Ar flow, and the substrate choice (e.g., freshly cleaved graphite or an Au-coated surface) influences the preferred crystal orientation. The SnTe sheets achieved are high-quality single crystals, as evidenced by the sharp angle-resolved Raman modes and distinct facets, and are typically a few tens of nanometers thick with lateral dimensions of several micrometers. Field-effect transistors made from these CVD-grown SnTe flakes show p-type behavior, and SnTe photodetectors exhibit a strong infrared response, underscoring the material’s potential as a topological crystalline insulator and IR-active 2D semiconductor.

Table 1 outlines, for each material, the specific sintering parameters, including precursor selection, temperature, and pressure conditions, along with the resulting nanosheet thickness and lateral dimensions. Reported applications of each synthesized material are also provided.

### 2.2. Top-Down Techniques: Liquid-Phase and Mechanical Exfoliation

LPE is a widely used technique for producing 2D group IV monochalcogenides in solution, enabling scalable synthesis of ultrathin nanosheets. This method involves the dispersion of bulk crystal powders in suitable solvents, followed by ultrasonic agitation to break the weak van der Waals interactions between adjacent layers. Solvent selection plays an important role in stabilizing exfoliated flakes, with commonly used solvents including *N*-methyl-2-pyrrolidone (NMP), isopropanol, and ethanol. The exfoliation yield and flake thickness can be tuned through sonication time, temperature, and centrifugation speed, with prolonged sonication yielding thinner nanosheets but at the cost of reduced lateral dimensions. The resulting 2D materials exhibit varying thickness distributions, with lateral sizes typically ranging from tens to hundreds of nanometers. While LPE provides high-throughput production of ultrathin 2D monochalcogenides, challenges remain in achieving precise thickness control and large-area flakes, necessitating post-processing techniques such as size-selective centrifugation to improve material uniformity.

Figure 2 illustrates the primary mechanisms involved in LPE of layered materials: ion intercalation, ion exchange, and sonication-assisted exfoliation. In the ion intercalation method (Figure 2a), guest ions (represented as green spheres) are introduced into the layered structure within a liquid medium. These ions penetrate between the layers, expanding the interlayer spacing and weakening van der Waals forces. Subsequent agitation—such as shear mixing, ultrasonication, or thermal activation—further disrupts interlayer bonding, leading to the separation of individual nanosheets and forming a stable dispersion. In the ion exchange mechanism (Figure 2b), some layered materials naturally contain intercalated ions that help maintain charge neutrality. By immersing these materials in a liquid environment, the native ions (depicted as red spheres) can be exchanged for larger foreign ions (green spheres). This process modifies the interlayer interactions, facilitating exfoliation upon agitation and resulting in a dispersed suspension of nanosheets. In sonication-assisted exfoliation (Figure 2c), layered materials are immersed in a suitable solvent and subjected to ultrasonication or shear mixing. The mechanical energy disrupts interlayer interactions, promoting the formation of thin nanosheets. The choice of solvent is important as it determines the stability of the dispersion—solvents with optimal surface energy prevent reaggregation, while those with poor compatibility lead to sedimentation. This mechanism is widely applied to produce high-quality dispersions of various 2D materials, including graphene oxide in polar solvents like water.

Mechanical exfoliation is a top-down approach that relies on the direct cleavage of layered bulk crystals to produce high-quality, ultrathin 2D materials. This method, commonly known as the “Scotch tape method”, involves repeatedly peeling a bulk crystal until monolayer or few-layer nanosheets are obtained [49]. Compared with LPE, mechanical exfoliation typically yields flakes with higher crystallinity, fewer defects, and larger lateral sizes, often on the order of tens of micrometers. However, the technique suffers from low scalability and limited control over thickness distribution. Materials with non-layered (3D) bonding structures, such as GeTe and SnTe, are difficult to exfoliate mechanically due to their lack of van der Waals interlayer forces. Despite its limitations, mechanical exfoliation remains the preferred method for producing high-quality 2D monochalcogenides for fundamental studies as the flakes preserve the intrinsic electronic and optical properties of the bulk material with minimal structural degradation. Although there have been a few attempts to automate the process, they all fall short at some stage of production [50,51,52].

The following sections provide a detailed overview of recent advancements in the synthesis of GeS, GeSe, GeTe, SnS, SnSe, and SnTe via liquid-phase exfoliation and mechanical exfoliation, highlighting the key parameters that influence exfoliation efficiency, including solvent selection, sonication conditions, and centrifugation techniques for LPE, as well as exfoliation methods such as Scotch tape and gold-assisted peeling for ME. Additionally, the discussion emphasizes the quality of the resulting nanosheets in terms of thickness, lateral size, and crystallinity.

#### 2.2.1. GeS

Recent work shows that GeS crystals can be exfoliated in organic solvents under ambient conditions. For example, sonication in NMP yields stable dispersions of GeS nanosheets. The exfoliated GeS is obtained as multilayer flakes (a few nanometers thick), and careful analysis indicates good crystallinity with oxidation only slowly attacking sheet edges over days, leaving the basal planes intact [53]. GeS can also be mechanically cleaved from bulk crystals (often grown by chemical vapor transport) using methods like the Scotch tape technique. Tan et al. demonstrated thin GeS flakes ranging from ~65 nm down to ~8 nm in thickness via micromechanical exfoliation [54]. These flakes are highly crystalline and typically exhibit lateral dimensions on the order of microns, making them suitable for device fabrication despite their thicker few-layer nature compared with the LPE samples.

#### 2.2.2. GeSe

Bianca et al. used anhydrous isopropanol to exfoliate GeSe, obtaining a colloidal suspension of predominantly few-layer (≤5 layers) GeSe nanoflakes [55]. AFM revealed that while occasional monolayers and thicker multilayers were present, the majority of flakes were two to five layers thick. These nanosheets had relatively small lateral sizes (15–180 nm, peaking around ~36 nm), reflecting the trade-off of LPE: extremely thin sheets but with limited lateral extent. Notably, the nanosheets retained the orthorhombic crystal structure of bulk GeSe with minimal degradation, as evidenced by intact lattice fringes and selected-area electron diffraction. GeSe is a layered compound and can be cleaved to yield larger-area flakes. Ma et al. [56] also obtained few-layer nanoflakes with lateral sizes up to 300 nm. Several groups have exfoliated GeSe by mechanical cleavage of melt-grown or vapor-transport-grown crystals. For instance, Yap et al. achieved GeSe flakes as thin as ~14 nm using standard micromechanical exfoliation [57]. Typically, mechanically exfoliated GeSe flakes are on the order of tens of nanometers thick (e.g., 30–120 nm is common), but they offer relatively large lateral dimensions (often many microns across) and high crystalline quality. Such flakes have been used to study GeSe’s anisotropic optical and electronic properties since they preserve the pristine structure of the bulk with low defect densities.

#### 2.2.3. GeTe

Although GeTe is not a classical van der Waals layered solid, it has been successfully exfoliated into 2D form due to the low exfoliation energy required to peel off one monolayer of rhombohedral GeTe as computed to be approximately 0.63 J/m^2^ [58,59]. Zhang et al. demonstrated sonication-assisted LPE of rhombohedral GeTe in ethanol, producing dispersions of GeTe nanosheets [60]. By tuning sonication and centrifugation parameters, they obtained few-layer GeTe sheets predominantly two to four layers thick and even observed occasional monolayer flakes. The exfoliated GeTe was found to be high quality and could be enriched in thin layers by sequential centrifugation, enabling optical studies that showed an evolution of bandgap with thickness. These ultrathin GeTe nanosheets also exhibited novel sensing capabilities (e.g., selective Fe^3+^ detection) due to their high surface area and preserved crystallinity [61]. In contrast to GeS and GeSe, mechanically cleaving GeTe is extremely challenging. Bulk GeTe adopts a three-dimensional (non-van der Waals) crystal structure (rock salt derived) that does not readily shear into thin layers. To date, there have been no reports of obtaining 2D GeTe by conventional mechanical exfoliation.

#### 2.2.4. SnS

SnS is a strongly layered group IV monochalcogenide, but its high interlayer binding energy makes monolayer production difficult. Recent advances have overcome this barrier: Sarkar et al. introduced a thermally assisted LPE technique to isolate SnS monolayers [62,63]. By heating the SnS solvent mixture during sonication, they could weaken interlayer forces and achieve large-scale exfoliation of single-layer SnS. The resulting SnS monolayers were reported to be highly crystalline with lateral sizes on the order of a few hundred nanometers and exhibited enhanced optoelectronic properties compared with thicker flakes. Earlier LPE efforts using conventional sonication (e.g., in NMP) typically yielded thicker few-layer SnS (5–10 nm thick) and lower monolayer yields, but the new method significantly improves both the thickness (true monolayers) and quality of SnS nanosheets [64]. SnS flakes can be prepared by mechanical cleavage, and an interesting refinement involves using a gold film to assist exfoliation. Higashitarumizu et al. demonstrated that peeling SnS with an Au layer (Au-assisted exfoliation) produces much thinner flakes than tape alone [65]. In their study, SnS layers down to ~4.3 nm thick (only a few layers) were obtained. These ultrathin SnS flakes remained highly crystalline, with AFM measurements showing atomically smooth surfaces (~0.1 nm surface roughness). The lateral size of mechanically cleaved SnS is typically on the order of tens of microns, and such flakes have been used to build photodetectors and FETs. The main limitation of mechanical exfoliation for SnS is yield and thickness control, yet it consistently provides high-quality, large-area sheets suitable for prototype devices.

#### 2.2.5. SnSe

SnSe has been exfoliated in liquids to produce few-layer nanosheets for applications ranging from photodetectors to thermoelectrics. Huang et al. sonicated SnSe powder in isopropanol for ~20 h and used cascade centrifugation to sort flakes by thickness [66]. They reported mean SnSe thicknesses of ~4.3 nm (at high centrifuge speed), ~8.9 nm (at low speed), etc., demonstrating some control over layer count via centrifugation. Similarly, Ye et al. found that among various solvents, NMP was most effective for SnSe, yielding nanosheets as thin as ~2.5 nm (just a few monolayers) [67]. Ju and Kim [68,69] prepared nanosheets by LPE and lithium intercalation, resulting in further improvement of yield. The obtained nanoflakes had a thickness of around six monolayers but with lateral sizes of around 300 nm, achieving a higher surface area-to-thickness ratio compared to the lateral sizes of only 50 nm in [66]. In these LPE approaches, the flakes tend to be small (roughly 100–300 nm lateral size) but exhibit good structural integrity and could be obtained in gram-scale quantities. However, Tayari et al. obtained nanoflakes with tens of microns lateral size yet a thickness of around 100 nm on Si/SiO_2_ substrates [70]. Being a van der Waals solid, SnSe can be mechanically exfoliated into thin sheets, although achieving monolayers is rare, due to the fact that SnSe exhibits a strong inter-layer coupling [71]. Several studies have demonstrated few-layer SnSe via Scotch tape exfoliation of SnSe crystals. For example, Yang et al. reported SnSe flake thicknesses down to ~7 nm (alongside thicker ~20–30 nm flakes) when exfoliating CVT-grown crystals [72]. Other groups obtained SnSe flakes in the 50–100 nm thickness range by simple cleavage [73,74,75,76]. In general, mechanically exfoliated SnSe maintains high crystal quality and low defect densities, and the flakes are sizable (lateral dimensions of tens of microns are common). The challenge, however, is that controlling the layer number below ~10 nm via mechanical means is difficult, so LPE or chemical intercalation methods are often preferred when truly ultrathin (≈1–2 nm) SnSe is required.

#### 2.2.6. SnTe

Despite SnTe’s 3D bonding, Singh et al. synthesized 2D SnTe by liquid-phase exfoliation using isopropanol as the solvent [77]. In practice, bulk SnTe ingots were first powdered and then sonicated in IPA for several hours. This process yielded dispersions of SnTe nanosheets, which, when deposited, formed few-layer films. The obtained SnTe sheets were not strictly monolayer but were thin enough to show significantly altered properties: for instance, exfoliated SnTe exhibited an enhanced room-temperature thermoelectric figure of merit (~0.17) compared with bulk SnTe (~0.005). This performance increase indicates the presence of nanometer-thick SnTe with increased Seebeck coefficient and reduced thermal conductivity, consistent with few-layer behavior. The LPE conditions (IPA solvent and prolonged sonication) thus allow SnTe to be divided into thin platelets that retain the rock salt structure but with thickness confinement. By using ultrasonication-assisted liquid-phase exfoliation, Qiao et al. [78] synthesized SnTe quantum dots with thicknesses of around 10–15 nm. Direct mechanical exfoliation of SnTe has not been successful. Because SnTe crystallizes in a robust rock salt structure (with no natural van der Waals gaps), it resists cleavage into atomic layers.

Table 2 shows, for each material, the specific exfoliation parameters used in LPE and ME, including solvent selection, sonication conditions, and exfoliation methods. The resulting nanosheet thickness, lateral size, and crystallinity are also detailed. Additionally, the table includes reported applications of each synthesized material.

## 3. Structure

Among the most common crystal structures found in 2D group IV monochalcogenide semiconductors (GeS, GeSe, SnS, and SnSe) are the cubic NaCl-like and orthorhombic phosphorous-like configurations. Figure 3 presents the side and top views of these structures along with their Brillouin zone representation. At low temperatures, these materials exhibit a stable orthorhombic α-phase (Pnma), characterized by a distorted rock salt structure comprising zigzag double-layer planes that are held together by van der Waals (vdW) interactions [80,81,82]. Within this configuration, Ge^2+^ or Sn^2+^ cations are coordinated within a distorted octahedral arrangement, where the cation–anion bond angles deviate slightly due to the presence of three shorter and three longer bonds. This distortion is attributed to the lone-pair effects of Sn(5s) and Ge(4s) orbitals, influencing the overall structural arrangement [83,84].

For SnS, the α-phase is its most stable structural form, with lattice constants of a = 11.14 Å, b = 3.97 Å, and c = 4.34 Å [23]. When heated beyond 873 K, it undergoes a phase transition to the β-phase, where the lattice parameters change to a = 4.12 Å, b = 11.48 Å, and c = 4.17 Å [23]. A similar temperature-dependent phase evolution is observed in SnSe, which follows a comparable transition pattern along different crystallographic directions. The thermodynamically stable α-phase of SnSe is characterized by Pnma symmetry, with lattice constants of a = 11.49 Å, b = 4.44 Å, and c = 4.14 Å. However, α-SnSe exhibits metallic transport behavior from 300 to 525 K, transitioning to thermally activated semiconducting behavior above 800 K. This phase transition from the α structure to the β structure (a = 4.31 Å, b = 11.70 Å, c = 4.31 Å) is linked to thermal excitation and structural changes [85,86,87,88].

SnTe exhibits three distinct crystalline phases, each with unique structural characteristics. The low-temperature phase (α-SnTe) has a rhombohedral structure with lattice parameters a = 6.325 Å, α = 89.895°, and belongs to the R3m space group. The β-phase, which is the most stable at room temperature, adopts a rock salt cubic structure with lattice parameters a = 6.3268 Å, α = 90°, and belongs to the Fm-3m space group. Last, the γ-phase has an orthorhombic crystal structure with lattice parameters a = 11.95 Å, b = 4.37 Å, and c = 4.48 Å, classified under the Pnma space group. The β-SnTe phase, which is thermodynamically stable above 100 K, can transition to α-SnTe at temperatures below 100 K due to a structural distortion along the [89] direction. This transformation is driven by the instability of the cubic lattice, causing a rearrangement of atomic positions, which affects symmetry and electronic properties. Additionally, applying an external pressure exceeding 18 kbar induces a transformation from β-SnTe to the orthorhombic γ-SnTe phase. Given that α-SnTe is only stable at cryogenic temperatures (<100 K) and γ-SnTe requires high pressure (>18 kbar) to form, the β-phase remains the most studied and is commonly referred to as SnTe in research due to its stability under ambient conditions.

At room temperature, GeS adopts the orthorhombic α-phase as its thermodynamically stable structure, with lattice parameters of a = 10.47 Å, b = 3.64 Å, and c = 4.30 Å [23]. When heated to 600 K, GeS undergoes a phase transition to a hexagonal β-GeS phase, which has lattice constants of a = b = 8.70 Å and c = 8.73 Å [12]. At even higher temperatures, exceeding 873 K, GeS transitions into another β-phase, with lattice parameters a = 4.24 Å, b = 10.45 Å, and c = 3.69 Å [90]. Similarly, GeSe exhibits an α-phase as its preferred thermodynamic structure, characterized by lattice constants of a = 4.40 Å, b = 3.85 Å, and c = 10.82 Å [23]. As the temperature rises, α-GeSe undergoes a transformation into a β1 phase, where its Ge-Se six-membered rings shift from their usual chair conformation to a boat-like arrangement. Single-crystal X-ray diffraction data confirm that β1-GeSe crystallizes in the orthorhombic space group Pnma, with lattice parameters a = 8.09 Å, b = 3.82 Å, and c = 5.81 Å [91]. The detailed structural parameters for these phases are summarized in Table 3.

**Table 3 materials-18-01530-t003:** Crystal structure parameters of group IV monochalcogenides.

	Space Group	Crystal Structure	Lattice Parameters (A)	References
GeS	$D_{2h}^{16}$	Orthorhombic	a = 4.3; b = 10.47; c = 3.65	[11]
GeSe	$D_{2h}^{16}$	Orthorhombic	a = 10.84; b = 3.83; c = 4.39	[12,92]
GeTe	R3m	Rhombohedral	a = b = 4.17 Å, and c = 10.71	[93]
SnS	$D_{2h}^{16}$	Orthorhombic	a = 4.33; b = 11.19; c = 3.98	[5]
SnSe	$D_{2h}^{16}$	Orthorhombic	a = 11.49; b = 4.15; c = 4.44	[13,94]
SnTe	Fm-3m	Cubic	a = 6.04 Å, α = 90°	[47]

These structural transformations between the different phases induce significant changes in the electronic properties of 2D group IV monochalcogenides. For example, phase transitions can alter the band gap from indirect to direct, affecting optical absorption and carrier mobility. Additionally, such transitions influence conductivity, thermoelectric performance, and anisotropic electronic behavior, which are important for applications.

**Figure 3 materials-18-01530-f003:**
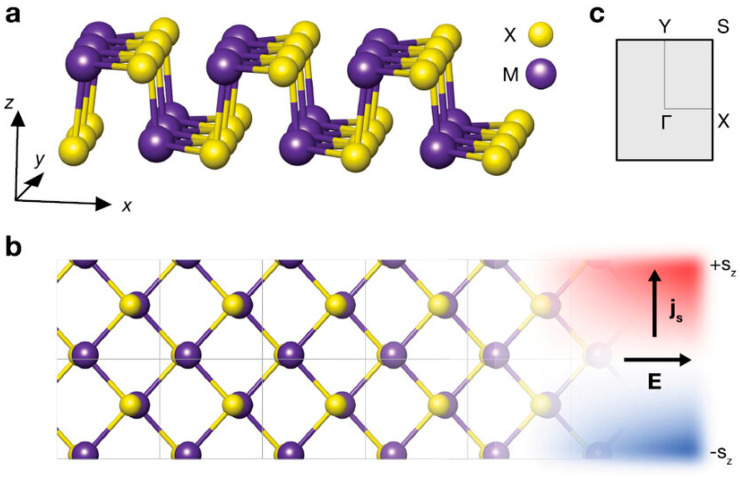
Structure of 2D group IV monochalcogenides. (**a**,**b**) The side and top views, respectively, and (**c**) the Brillouin zone scheme. Reproduced with permission from [95].

GeTe shares a structural resemblance to α-As, where Ge and Te atoms alternately replace As atoms within the As layers, forming a layered structure. A GeTe unit cell consists of three Ge and three Te atoms, arranged in three buckled GeTe layers stacked in an ABC sequence. The lattice constants for bulk GeTe, calculated using PBE-D2 theory, were found to be a = b = 4.15 Å and c = 10.70 Å, which aligned closely with experimental values (a = b = 4.17 Å, c = 10.71 Å) [93]. Within a single GeTe layer, each Ge atom bonds with three neighboring Te atoms, and vice versa, resulting in the formation of a 2D hexagonal lattice. The Ge–Te bond distances within the same layer (intralayer) and between adjacent layers (interlayer) are 2.83 Å and 3.15 Å, respectively, while the buckling height of the layer is approximately 1.51 Å. After undergoing full atomic relaxation, the monolayer GeTe structure remains similar to its bulk counterpart, maintaining its layered nature. However, the monolayer form of GeTe experiences a noticeable lattice contraction, with optimized parameters of a = b = 3.96 Å. This lattice shrinkage leads to a higher buckling height (1.57 Å) and a reduced Ge–Te bond length (2.77 Å), reinforcing its structural similarity to arsenene and blue phosphorene.

## 4. Electronic Properties

Understanding the electronic band structure of 2D materials is important for analyzing their electronic and optical behaviors, particularly in electronic and photonic device applications. One of the main attributes of group IV monochalcogenides is their highly tunable bandgap, which can be adjusted within the visible spectrum, making them adaptable for various optoelectronic applications. Experimental studies report that the bandgap in bulk form varies from approximately 0.8 eV (SnSe) to 2.0 eV (GeS). To obtain precise band structure predictions, advanced beyond-DFT methods are often employed as they provide a more accurate treatment of electron exchange and correlation effects, overcoming the limitations of standard DFT approaches [96,97]. The electronic band structures of monolayer, bilayer, and bulk group IV monochalcogenides have been computed in prior theoretical studies [80], with similar findings reported in other computational works [98,99]. Figure 4 presents these band structures, while Table 4 provides a comparison of bandgap values obtained from various theoretical and experimental studies. Due to their orthorhombic symmetry, group IV monochalcogenides possess a rectangular Brillouin zone in two-dimensional form, with high-symmetry points Γ, X, Y, and T in the kx–ky plane for monolayers and bilayers. In bulk form, additional high-symmetry points emerge along the kz direction, reflecting the three-dimensional nature of the electronic structure.

**Table 4 materials-18-01530-t004:** Bandgaps for monolayer, bilayer, and bulk group IV monochalcogenides. All values are given in eV.

	Source	Monolayer	Bilayer	Bulk	References
GeS	Theory	2.32	2.20	1.81	[80]
		1.73			[99]
		1.66			[100]
GeSe	Theory	1.54	1.45	1.07	[80]
		1.73			[99]
		1.66			[100]
GeTe	Theory	2.35		0.65	[59]
SnS	Theory	1.96	1.60	1.24	[80]
		2.14			[99]
		2.24			[100]
		2.57			[98]
SnSe	Theory	1.40	1.20	1	[80]
		1.51			[99]
		1.39			[100]
SnTe	Theory			0.20	[101]

For the monolayers, each unit cell contains 20 electrons—4 from the group IV element and 6 from the chalcogen—which occupy ten spin-degenerate electronic bands. For the bilayers and bulk, we have twice as many atoms; therefore, a total of 40 electrons are accommodated in 20 bands.

Several common features can be observed in the electronic band structures of group IV monochalcogenides. In their bulk form, these materials exhibit indirect bandgaps, with the valence band maximum (VBM) positioned along the Γ–X direction. The conduction band minimum (CBM) location varies between Γ–X for Sn-based compounds and near Γ for Ge-based compounds. The bandgap (E_g_) trend follows the order E_g_(SnSe) < E_g_(GeSe) < E_g_(SnS) < E_g_(GeS), a relationship that holds consistently across monolayers, bilayers, and bulk forms.

A well-known characteristic of group IV monochalcogenides is the bandgap’s strong dependence on the number of layers, a phenomenon observed in several layered materials. This tunability arises from multiple factors, including orbital contributions at the band edges and quantum confinement effects [102]. Experimentally, Brent et al. demonstrated the bandgap tunability of SnS, reporting a reduction from 1.65 eV in bulk to 1.03 eV in bilayer form [103]. This tunability is significant because it enables the material’s optical and electronic properties to be tailored for applications where precise control over the bandgap can optimize device performance.

In the bilayer form, group IV monochalcogenides exhibit notable differences from their bulk counterparts, primarily in the form of larger bandgap values. While SnS, SnSe, and GeS retain their indirect bandgap nature with only minor variations in the VBM and CBM positions, GeSe undergoes a transition to a direct bandgap semiconductor. This shift is primarily caused by modifications in the conduction band, where the CBM moves from the Γ point to the valley along the Γ–X direction. Experimental validation of this indirect-to-direct transition in few-layer and monolayer GeSe was recently reported by Zhao et al., confirming DFT-based predictions that this transition occurs when the layer count is reduced from four to three [104].

In their monolayer form, group IV monochalcogenides exhibit the largest bandgap values, which range from approximately 1.4 eV in SnSe to 2.4 eV in GeS. Several theoretical studies indicate that most of these materials maintain indirect bandgaps, with VBM and CBM positions similar to those in bilayer and bulk forms. However, additional local extrema appear in the valence and conduction bands, positioned very close to the band edges with energy differences of only a few meV [80]. As a result, materials such as SnS, SnSe, and GeS exhibit competing direct and indirect bandgaps, where the direct transitions are only slightly higher in energy than their indirect counterparts. These small energy differences contribute to variability in reported band structures across different theoretical studies.

Another remarkable property of group IV monochalcogenides is the presence of two distinct pairs of valleys in the valence and conduction bands, typically found along the Γ–X and Γ–Y directions. This valley structure enables selective excitation using linearly polarized light, offering potential control over both electronic and spin states. Because each valley pair can be optically activated independently using polarized excitation, these materials show significant promise for spintronics and valleytronics applications.

Figure 5 illustrates the structural transitions in GeTe and SnTe, where both materials exhibit a stable rhombohedral α-phase below their respective transition temperatures Tc, 670 K for GeTe and 98 K for SnTe. GeTe and SnTe experience a symmetry-breaking displacement along the cubic [89] direction, shifting their cation and anion sublattices. This displacement results in a relative shift (τ) of 0.028 for GeTe and 0.014 for SnTe, influencing their electronic properties. The structural transition significantly impacts the electronic band structure, with α-GeTe displaying a direct band gap of 0.59 eV at the L point, compared with 0.17 eV in its cubic phase. In α-SnTe, the strong cation–anion interaction suppresses band inversion, leading to a normal semiconductor behavior with a band gap of 0.20 eV. These structural and electronic modifications highlight the critical role of phase transitions in tuning the properties of GeTe and SnTe for electronic and optoelectronic applications [105].

## 5. Characterization

The characterization of group IV monochalcogenides encounters major obstacles due to their structural anisotropy, environmental sensitivity, and intrinsic material properties. Mechanical exfoliation, while widely used to produce monolayers, often results in inhomogeneous flake thickness and aggregation due to strong interlayer van der Waals forces. This aggregation obscures monolayer-specific properties during spectroscopic analysis. For instance, Raman spectra of aggregated SnS exhibit peak broadening (FWHM > 15 cm^−1^), masking the layer-dependent phonon mode shifts essential for thickness determination. Liquid exfoliation methods using ionic liquids or surfactants can reduce aggregation but introduce surface adsorbates that alter electronic properties [106]. Group IV MCs degrade rapidly under ambient conditions due to hydrolysis and oxidation. The polar nature of monolayers (e.g., SnSe) facilitates water molecule adsorption, leading to 80% optical contrast loss within 24 h. This instability necessitates glove-box-based preparation and in situ characterization, which are incompatible with many analytical tools requiring ambient exposure. Even brief air exposure (<5 min) during transfer to TEM chambers induces measurable oxidation, as evidenced by XPS-detectable SnO_2_ and GeO_2_ formation [107]. While effective for thickness estimation in ideal conditions, Raman struggles with defect-rich group IV MCs. The anisotropic phonon modes in GeS exhibit <5 cm^−1^ shifts between monolayers and bilayers, requiring sub-μm spatial resolution rarely achieved in standard systems. Defect-induced peak splitting further complicates interpretation [108]. Atomic-resolution TEM faces challenges due to beam sensitivity; monolayer SnSe undergoes amorphization at electron doses > 50 e−/Å^2^, limiting observation windows. Low-dose techniques sacrifice signal-to-noise ratios, obscuring defects like sulfur vacancies (Vs) that occur at densities > 10^12^ cm^−2^ in CVD-grown SnS [106]. Surface roughness artifacts from substrate interactions (e.g., SiO_2_-induced rippling) produce ±0.3 nm height measurement errors, making monolayer discrimination unreliable. Conductive AFM measurements of SnSe show 50% variability in current mapping due to contact resistance fluctuations [108]. Many predicted properties (e.g., valley polarization in GeSe) require measurement under ultrahigh vacuum (UHV) to minimize surface adsorbate effects. However, most optoelectronic characterization (photoluminescence and photoconductivity) occurs in ambient conditions. For example, SnS monolayers exhibit theoretically predicted photoluminescence quantum yields of 15% in UHV but <0.1% in air due to trap state formation [109]. DFT simulations often assume pristine monolayers, neglecting ubiquitous defects like chalcogen vacancies (V_S_ and V_Se_) and grain boundaries. Experimental STEM-EELS data show Vs concentrations up to 3% in MBE-grown SnSe, reducing theoretical bandgaps from 1.8 eV to 1.2 eV. Machine-learning-assisted simulations are now bridging this gap by incorporating defect statistics into property predictions [106].

### 5.1. Morphological and Structural Characterization

The following sections provide a detailed overview of recent advancements in the characterization of GeS, GeSe, GeTe, SnS, SnSe, and SnTe, highlighting the morphological and structural characteristics of the obtained materials.

#### 5.1.1. GeS

Tan et al. [54] investigated the anisotropic optical properties and thickness-dependent behavior of 2D layered germanium sulfide (GeS). Their study identified three distinct Raman-scattering peaks, B_3g_, A^1^_g_, and A^2^_g_ modes, which exhibited strong polarization dependence. These findings confirm that polarized Raman spectroscopy serves as a reliable technique for determining the crystal orientation of anisotropic layered GeS. Additionally, photoluminescence (PL) measurements revealed an optical transition peak near 1.66 eV at room temperature, which corresponded to a direct bandgap transition in GeS. The study also presented polarization-dependent PL characteristics and anisotropic absorption, providing strong evidence of directional optical transitions near the band edge. These results underscore the significant anisotropic optical properties of GeS, making it a promising material for polarization-sensitive optoelectronic applications.

Recent advancements in the synthesis of GeS have led to the development of large, high-quality single-crystal flakes using a space-confined APCVD method [105]. This technique enables the growth of well-crystallized, orthorhombic GeS flakes with a lateral length of up to 100 μm and a thickness of approximately 52 nm. The growth process is highly dependent on time, with flake size increasing as deposition progresses.

Figure 6 provides a structural characterization of GeS synthesized using space-confined APCVD. Atomic force microscopy (AFM) imaging confirms a flake thickness of 52 nm, while scanning electron microscopy (SEM) reveals a smooth surface with well-defined edges. High-resolution transmission electron microscopy (HRTEM) imaging displays a well-ordered atomic arrangement with a lattice spacing of 0.278 nm along the [011] crystal phase, supported by the selected area electron diffraction (SAED) pattern, which confirms the high crystallinity of single-crystal GeS. X-ray diffraction (XRD) analysis indicates that the experimental samples align with the standard orthorhombic GeS crystal structure (space group Pnam), yielding lattice constants of a = 10.444 Å, b = 4.277 Å, and c = 3.633 Å, closely matching reference values. Raman spectroscopy reveals characteristic peaks at 210 cm^−1^, 233 cm^−1^, and 267 cm^−1^, corresponding to in-plane shear vibrations of neighboring layers in the armchair and zigzag directions. Elemental composition analysis using energy dispersive spectroscopy (EDS) confirms a Ge-to-S atomic ratio of 48.5:51.5, demonstrating stoichiometric consistency in the synthesized GeS flakes.

#### 5.1.2. GeSe

Yumigeta et al. [37] successfully synthesized layered anisotropic GeSe semiconductors along with the GeSe_2_ phase using a low-temperature (~400 °C) and atmospheric pressure chemical vapor deposition CVD approach. By utilizing halide-based precursors, they demonstrated that GeI_2_ and H_2_Se react in the gas phase, leading to nucleation on various substrates such as sapphire, Ge, GaAs, and highly oriented pyrolytic graphite (HOPG), as depicted in Figure 7a. Following the nucleation process, layer-by-layer growth was observed, enabling the formation of anisotropic layered materials with controlled structural properties.

Growth experiments conducted at 420 °C on germanium substrates resulted in the formation of rectangular-shaped ultrathin GeSe flakes, with lateral dimensions extending to tens of microns (Figure 7b) [37]. The nature of the substrate surface plays a crucial role in determining the growth mode. For instance, truncated rectangular pyramidal crystals were observed when using single-crystal germanium substrates (Figure 7g), whereas flat and thin GeSe sheets formed on HOPG surfaces (Figure 7e). On GaAs substrates, however, the GeSe growth exhibited out-of-plane flower-like features (Figure 7f,g). This variation in growth morphology may be attributed to differences in elastic energy, which was calculated as 0.032 meV on Ge (100) substrates and 0.046 meV on GaAs. The relatively higher elastic energy on GaAs could be responsible for promoting out-of-plane crystal growth.

#### 5.1.3. GeTe

GeTe is generally synthesized using methods other than CVD, leading to limited reports of CVD-grown GeTe [110]. Yao et al. [111] performed a one-step CVD method to simultaneously grow two types of nanosheets. Importantly, the nanosheets can be completely separated by selecting the deposition region as different regions within the reactor undergo distinct temperature and pressure conditions, which influence the nucleation and growth of specific phases. In detail, the controllable and completely separable growth for α-GeTe and Te nanosheets was realized by using one precursor-GeTe powder through an APCVD approach. The as-grown α-GeTe nanosheets exhibited a typical triangular shape with edge lengths of 24 microns and a thickness of 7.1 nm, as measured by AFM. Te-rich zones, facilitated by the high saturated vapor pressure and low melting point of tellurium, were synthesized on mica substrates located in low-temperature zones. These Te nanosheets had a thickness of 8.5 nm. Thickness measurements are critical for evaluating the material’s electronic and optical properties and ensuring reproducibility in synthesis. High-resolution XRD confirmed the crystal quality of the nanosheets, with Raman spectra revealing characteristic peaks at 86 and 123.5 cm^−1^, corresponding to E_g_ and A_g_ vibration modes. HRTEM identified a lattice plane spacing of 0.207 nm for the (110) plane, with SAED patterns confirming the single-crystal nature of the α-GeTe nanosheets. EDS analysis showed a 1:1 atomic ratio, and elemental mapping indicated a uniform distribution across the nanosheets.

Another study investigated the synthesis and characterization of GeTe nanosheets obtained through an LPE method [112]. The synthesis process involved exfoliating bulk GeTe in deionized water using lithium hydroxide-assisted ion intercalation, followed by ultrasonication and purification (Figure 8a). AFM measurements confirmed a flake thickness of approximately 4.85 nm, corresponding to seven to eight layers (Figure 8b). Raman spectroscopy identified characteristic Ge–Te and Te–Te vibrational modes at 121 cm^−1^ and 141 cm^−1^, with observed blue and red shifts due to variations in interlayer interactions (Figure 8c). HRTEM images revealed the high crystalline quality of the nanosheets, with selected area electron diffraction confirming a single-crystal rhombohedral structure (Figure 8d).

#### 5.1.4. SnS

Tian et al. [64] successfully synthesized SnS nanoflakes using CVD, with the results shown in Figure 9. A high density of 2D SnS nanoplates was grown on mica substrates, as revealed by optical microscopy images (Figure 9b). These nanoplates exhibit a rhombic morphology, with lateral dimensions ranging from 5 to 15 μm, and well-defined facets, indicative of their single-crystalline nature. Unlike SiO_2_/Si-supported 2D materials, which display thickness-dependent interference colors, SnS nanoplates grown on mica exhibit low contrast under reflected light microscopy, making thickness determination challenging. AFM measurements confirmed that the thinnest synthesized SnS nanoplates had a minimum thickness of approximately 6 nm (~10 layers) (Figure 9d). Additionally, thicker nanoplates tend to exhibit sharper, well-defined edges, suggesting higher crystallinity (Figure 9c). A high-resolution TEM (HRTEM) image of a representative nanoplate (Figure 9e) and the corresponding selected-area electron diffraction (SAED) pattern (Figure 9f) confirmed that the lattice fringes exhibit a perfect rhombic arrangement, with sharp SAED spots, verifying the single-crystalline quality of the SnS nanoplates. Notably, the lattice spacing derived from SAED matches bulk SnS, confirming no structural alterations between bulk and few-layer forms, ensuring the retention of its intrinsic electronic and optical properties, which are crucial for device applications.

Raman spectroscopy analysis was also conducted to assess crystal quality across nanoplates of different thicknesses (6.3, 9.1, and 13.5 nm; Figure 9h). The observed Raman peaks (A_g_ modes at 40.0, 97.2, 191.9, and 218.7 cm^−1^; B_3g_ modes at 49.4 and 163.2 cm^−1^) are in close agreement with those in bulk SnS, confirming structural consistency. Notably, thicker nanoplates exhibit stronger and sharper Raman peaks, further reinforcing their higher crystallinity, a finding consistent with their well-defined edge morphologies observed via AFM.

#### 5.1.5. SnSe

Zhao et al. [45] reported a vapor transport deposition method for synthesizing single-crystal SnSe nanoplates on mica substrates. The synthesized nanoplates exhibited well-defined square shapes with lateral dimensions of 5–6 microns and a thickness of approximately 16 nm, confirmed by AFM. Transmission electron microscopy analysis indicated that the nanoplates maintained structural stability under electron beam irradiation, with their thin nature allowing for significant flexibility, enabling bending up to 180° without fracturing. HRTEM images displayed a regular atomic arrangement with a lattice spacing of 0.30 nm, consistent with the (011) planes of orthorhombic SnSe. EDX element mapping confirmed uniform Sn and Se distribution, with a stoichiometric ratio of 1:1. Additionally, Raman spectroscopy revealed characteristic phonon modes at 70.0, 105.5, 127.7, and 148.2 cm^−1^, corresponding to B_3g_ and A_g_ vibrational modes, which provided insights into interlayer coupling and structural quality.

High-quality SnSe nanoflakes with thicknesses below 100 nm and oriented along the [113] crystal axis were synthesized via physical vapor transport (PVT) at atmospheric pressure by Buruiana et al. [13]. The obtained nanoflakes exhibited well-defined rectangular shapes and uniform contrast, indicating a constant thickness across the flakes as shown in Figure 10a. AFM measurements confirmed that the nanoflakes had a controllable thickness below 100 nm (Figure 10b). EDX elemental mapping demonstrated a uniform distribution of Sn and Se, with a measured atomic ratio of approximately 47.46% Sn and 52.54% Se (Figure 10c). High-resolution transmission electron microscopy images revealed lattice fringes with a spacing of 3.04 Å, corresponding to reflections on the (011) planes, while selected area electron diffraction confirmed the single-crystalline orthorhombic nature of the SnSe nanoflakes (Figure 10d,e). Low-temperature Raman spectroscopy of SnSe nanoflakes (T = 5 K) showed six prominent Raman modes: A^1^_g_, B^1^_3g_, A^2^_g_, B^2^_3g_, A^3^_g_, and A^4^_g_, observed at 35.9 cm^−1^, 39.3 cm^−1^, 72.8 cm^−1^, 118.3 cm^−1^, 142.0 cm^−1^, and 154.1 cm^−1^, respectively (Figure 10f).

#### 5.1.6. SnTe

Liu et al. [114] demonstrated the controlled van der Waals growth of 2D SnTe nanoplates on mica substrates, highlighting their structural and optoelectronic properties. An SEM image of a transferred SnTe nanoplate onto a Si/SiO_2_ substrate (Figure 11a) reveals a near-square morphology. The AFM analysis (Figure 11b) confirmed that the nanoplates had a thickness of approximately 3.6 nm and a lateral size of 8 microns, corresponding to about six monolayers. The XRD pattern (Figure 11c) confirmed the crystalline nature of the SnTe nanoplates, while EDS elemental mapping images (Figure 11d–f) showed a uniform distribution of Sn and Te atoms. The Sn:Te atomic ratio is close to 1:1, consistent with the stoichiometric composition of SnTe.

The Raman spectrum, obtained using 785 nm laser excitation at room temperature (300 K), revealed three characteristic Raman peaks at 61 cm^−1^, 123 cm^−1^, and 139 cm^−1^ (Figure 11g). Optical transmission measurements performed on a 9 nm thick SnTe nanoplate on a SiO_2_ substrate (Figure 11h) showed a cutoff wavelength of 6.8 μm, corresponding to a bandgap of 0.197 eV, which defines the infrared detection limit of SnTe nanoplates. Further structural analysis using HRTEM (Figure 11i) identified the interplanar spacings of 0.315 nm and 0.224 nm, corresponding to the (200) and (220) crystal planes, respectively. These values align well with theoretical predictions and previously reported SnTe crystal structures. The selected-area electron diffraction (SAED) pattern (Figure 11j) confirmed the tetragonal crystal structure, reinforcing the high crystallinity and structural integrity of the synthesized SnTe nanoplates.

### 5.2. Raman Spectroscopy and Anisotropic Properties

The physical properties of monochalcogenides are linked to their crystal structures, which have a significant impact on their optical and electronic characteristics. They exhibit a diverse array of crystal phases, including orthorhombic, hexagonal, and cubic configurations, as shown in Table 3. These phase variations significantly influence the material’s electronic band structure, optical absorption, and mechanical properties, making. The specific phase is determined by variations in the oxidation states of the metal (in this case, Sn or Ge) and chalcogen atoms (S, Se, and Te). For instance, in Sn-based monochalcogenides, the chalcogen atom’s higher electronegativity compared with the metal results in the capture of two electrons from the Sn atom, causing a shift in its electronic configuration from 4d^10^5s^2^5p^2^ to 4d^10^5s^2^5p^0^. A similar effect is observed in Se-based monochalcogenides, where Se’s electronic configuration changes to 4s^2^4p^6^. Consequently, this electron redistribution distorts the buckled crystal layer structure, influencing electronic band alignment and optical absorption characteristics.

The unique physical properties of 2D materials often originate from the breaking of crystal structural symmetry. For instance, 2D graphene shows the highest symmetry, D_6h_ symmetry, with six-fold rotation in plane, six two-fold perpendicular axes, and a mirror plane. This structural arrangement leads to highly anisotropic optical, electronic, and thermal properties. However, group IV monochalcogenides consist of two elements with differing electronegativities, as opposed to the single-element composition of black phosphorene. Consequently, in odd layers of monochalcogenides, inversion symmetry is disrupted, leading to C_2*υ*_ symmetry, which includes not only a two-fold rotation but also two mirror planes. This unique feature provides even more extraordinary optical and electronic properties compared with phosphorene and TMDs. Beyond these intrinsic symmetries, external factors such as strain, substrate interactions, and doping can further fine-tune the electronic properties, enabling tunable optoelectronic responses and enhanced charge transport.

Their crystal structure of D_2_h^16^ in the Pnma space group exhibits eight atoms within each unit cell. At the Γ point, lattice vibrations can be described using the irreducible representation 4A_g_ + 2A_u_ + 2B_1g_ + 3B_1u_ + 4B_2g_+ B_2u_ + 2B_3g_ + 3B_3u_. Among these 21 vibrational modes, 12 are Raman active, making them useful for structural characterization, while 7 are infrared active and relevant for phonon-coupled electronic transitions, with 2 modes remaining optically inactive. According to the Raman polarization selection rules for the Pnma space group, four A_g_ and two B_3g_ phonon modes should be observable in the backscattering geometry along the Z crystallographic direction. These observed phonon modes are further categorized based on their irreducible representations within the $D_{2h}^{16}$ point group and assigned numbers corresponding to their increasing Raman shift, as presented in Table 5. Notably, GeTe and SnTe belong to different space groups, which results in distinct vibrational spectra and influences their phase stability and electronic interactions.

#### 5.2.1. GeS

Polarized Raman spectroscopy was performed on a 110 nm thick GeS flake under parallel (//) and cross-polarization (⊥) configurations, as shown in Figure 12 [115]. These measurements help reveal the anisotropic nature of phonon vibrations in GeS, highlighting its directional dependence on crystal orientation. The Raman-scattering intensities of all three Ag modes exhibit strong dependence on the polarization angle (Figure 12a). The corresponding polar plots of Raman intensities (Figure 12b–d) illustrate that the A^1^_g_ and A^2^_g_ modes exhibit periodicity with a 180° angle period and a 90° phase difference between them, while the B_3g_ mode shows periodic variations with a 90° angle period. A strong polarization dependence is also observed for the A_g_^3^ mode, suggesting enhanced sensitivity to strain and defects compared with other modes, further confirming the anisotropic vibrational properties of GeS.

#### 5.2.2. GeS

To assess the structural quality and anisotropic behavior of GeSe, angle-resolved Raman spectroscopy measurements were conducted, as shown in Figure 13 [37]. These measurements provide information into vibrational modes that reflect the degree of crystallinity and directional dependence of the material’s optical properties. The B^1^_3g_ mode at 153 cm^−1^ and the A^1^_g_ mode at 190 cm^−1^ exhibit four-fold symmetry due to their Raman tensor characteristics, which result in isotropic scattering in specific directions, preventing their use for crystal orientation determination. In contrast, the A^2^_g_ and A^3^_g_ modes display two-fold symmetry, which makes them effective for determining crystal orientation using optical techniques. The intensity of the A^2^_g_ and A^3^_g_ Raman modes reaches a maximum when the polarization vector angles are approximately 90° and 0°, respectively. The observed two-lobed features confirm the high crystallinity and pronounced anisotropy of the synthesized GeSe sheets.

#### 5.2.3. GeTe

Yang et al. [116] provided the first interpretation of vibrational modes in GeTe by combining experimental and theoretical Raman analysis, offering new insights into the phonon behavior of this material. The study was conducted on GeTe nanoflakes with thicknesses below 5 nm. According to group theory, only two Raman-active phonon modes are expected: the E mode at 96 cm^−1^ and the A_1_ mode at 121 cm^−1^, which play a crucial role in determining the vibrational and structural properties of GeTe.

Figure 14a presents the Raman spectrum measured on a GeTe nanoflake at room temperature (300 K) and at 80 K. Curve fitting at 300 K identified five peaks at 88.1 cm^−1^ (A), 124.6 cm^−1^ (B), 142.2 cm^−1^ (C), 158.4 cm^−1^ (D), and 225.5 cm^−1^ (E). While A and B are attributed to Ge–Te vibrations, the presence of additional peaks suggests complex phonon interactions beyond what is predicted by group theory. Peak C arises from long-range interactions within crystalline Te, while peak D is associated with the vibrational density of states of long disordered Te chains. Peak E is assigned to the antisymmetric stretching mode of the GeTe_4_ tetrahedra. Figure 14b illustrates the calculated phonon dispersion results.

The polar plots of Raman intensity for the two phonon modes are shown in Figure 14c–f. The E mode exhibits strong anisotropy with a 90° periodicity in both configurations, indicating significant directional dependence in phonon behavior, which can be leveraged for crystallographic orientation studies. The A_1_ mode, in contrast, displays a two-lobe pattern in the cross-polarization configuration, whereas in the parallel configuration, it presents an asymmetric response. The E mode serves as a reliable probe for determining crystal orientation, while the A_1_ mode provides valuable insights into temperature-dependent phase transitions, which are crucial for understanding phase stability and structural evolution in GeTe.

#### 5.2.4. SnS

Li et al. [117] demonstrated that for PVD-synthesized SnS flakes of varying thicknesses, Raman properties depend on the polarization configuration used during measurement. The Raman spectrum was collected in a backscattering geometry using a polarizer and an analyzer placed in the incident and scattered light paths, respectively. Two polarization configurations were adopted: parallel (∥), where the polarization directions of the incident and scattered light were aligned, and perpendicular (⊥), where they were orthogonal. These configurations allowed for the examination of directional anisotropies in vibrational modes, providing insights into phonon behavior. The polarization angle, defined as the angle between the polarization direction of the incident light and the armchair direction of the SnS flake, was tuned from 0° to 360° using a custom sample stage.

As shown in Figure 15, the Raman spectra show three strong peaks at 95.9 cm^−1^ (A_g_), 164.0 cm^−1^ (B_3g_), and 192.0 cm^−1^ (A_g_). The periodic intensity variation of A_g_ and B_3g_ modes with polarization angle confirms the anisotropic Raman response of SnS flakes. This effect is particularly evident in the polar plots of A_g_ (192.0 cm^−1^) and B_3g_ (164.0 cm^−1^), where the Raman intensity of A_g_ (192.0 cm^−1^, ∥) is greater along the armchair direction than along the zigzag direction. Notably, the intensity along the zigzag direction represents a secondary maximum rather than a minimum, differing from previous reports [5,64]. This discrepancy suggests possible variations in sample preparation, measurement conditions, or substrate effects, highlighting the sensitivity of Raman polarization studies to experimental parameters. For the B_3g_ mode (164.0 cm^−1^, ∥), the Raman intensity is lowest along the armchair and zigzag directions but reaches a maximum at 45° (225°) and 135° (315°), emphasizing the complex anisotropic vibrational properties of SnS.

#### 5.2.5. SnSe

The anisotropic structure of tin selenide nanoflakes was investigated by polarization-resolved Raman scattering by Buruiana et al. [13]. The analysis shown in Figure 16 revealed that the polarization axes of the Ag modes (A^1^_g_, A^2^_g_, A^3^_g_, and A^4^_g_) in the co-linear (∥) configuration align in the same direction (37°). In contrast, the polarization axes of the B3g modes (B_3g_^1^ and B_3g_^2^) are oriented at 81°, approximately 45° shifted from the Ag modes. The A_g_ modes display two-fold symmetry with an angle period of 180°, indicating in-plane anisotropy, while the B_3g_ modes exhibit four-fold symmetry with a 90° period, reflecting complex interlayer vibrational coupling. The observed Raman mode shapes in the ⊥ configuration resemble those of the same symmetry, with A_g_ modes displaying two-fold and B_3g_ modes displaying four-fold polarization patterns. Notably, the A^4^_g_ mode exhibits an unexpected three-fold symmetry instead of the anticipated two-fold symmetry.

The effect of temperature, ranging from 0 K to 300 K, on phonon-mode energies and linewidths was investigated using the phonon anharmonicity model developed by Balkanski et al. [118], which accounts for phonon–phonon interactions. The anharmonic coefficients, representing contributions from three- and four-phonon scattering processes, were compared with the harmonic frequencies at T = 0 K. The results indicate that three-phonon processes predominantly govern phonon decay channels as a function of temperature, affecting thermal conductivity and phonon lifetime.

#### 5.2.6. SnTe

Su et al. [47] studied the anisotropic behavior of SnTe nanosheets using angle-resolved polarized Raman scattering with a 532 nm laser in backscattering mode. Measurements were performed in both parallel and cross configurations to analyze polarization-dependent Raman responses. The results indicate that the intensity of the A_g_ mode remains constant with rotation, whereas the E_TO_ mode exhibits periodic intensity variations in both configurations.

A comparison between experimental data and theoretical predictions reveals discrepancies, highlighting potential influences from sample imperfections or measurement conditions. While the A_g_ mode intensity remains constant in the parallel configuration, theory suggests that no Raman signal should be observed in cross-polarization, which contradicts experimental results. This deviation is attributed to instrumental limitations. The E_TO_ mode’s periodic intensity variations align with theoretical expectations, reinforcing the presence of in-plane anisotropy in the tested square nanosheets.

Bulk SnTe, which has an isotropic cubic crystal structure, can experience symmetry changes when subjected to external influences such as doping or applied pressure. These modifications can affect electronic band structure and phonon interactions, leading to altered material properties. These factors can disrupt its inversion symmetry and alter its bonding force constants, leading to the appearance of new Raman peaks during phase transitions. A similar phenomenon is observed in MoS_2_, where growth-induced stress modifies symmetry. Since SnTe nanosheets are relatively thin, they are more susceptible to defects and stress-induced distortions. These factors can break symmetry, induce in-plane anisotropy, and significantly impact their vibrational and electronic properties.

Raman spectroscopy measurements on ultrathin SnTe films provide insights into their structural and electronic properties [119]. In particular, Raman-active transverse optical (TO) phonon modes are observed below the ferroelectric transition temperature. In a two-unit-cell-thick (UC) SnTe film, the TO mode at ~46.8 cm^−1^ is present at low temperatures and persists up to room temperature, shifting slightly with increasing temperature as shown in Figure 17. These results indicate enhanced ferroelectric stability in ultrathin SnTe layers, supporting the theoretical predictions of increased T_c_ in reduced dimensions.

## 6. Applications of Group IV Monochalcogenides

The unique structural and electronic properties of 2D group IV monochalcogenides have enabled their integration into a wide range of applications. These materials exhibit tunable bandgaps, high carrier mobility, and strong light absorption, making them highly promising for optoelectronic and photonic devices such as photodetectors, photovoltaics, and nonlinear optical components. Additionally, their anisotropic electronic properties and structural flexibility have facilitated their use in energy storage systems, including lithium-, sodium-, and potassium-ion batteries, as well as thermoelectric applications where optimized charge and heat transport characteristics are essential. The inherent chemical reactivity and defect engineering possibilities in these materials have also made them suitable for catalytic applications, including hydrogen evolution reactions and photocatalysis. Furthermore, their compatibility with flexible electronics and emerging neuromorphic computing architectures has positioned them as potential candidates for next-generation memory devices and artificial synapses.

### 6.1. GeS

Ulaganathan et al. [11] fabricated a broadband photodetector based on multilayered GeS with superior photoresponse, high stability, and fast response. The multilayered GeS-FETs exhibited a remarkably high photoresponsivity of R_λ_ ~206 A W^−1^ under 1.5 μW cm^−2^ illumination at λ = 633 nm, Vg = 0 V, and Vds = 10 V. The gate-dependent photoresponsivity was measured at R_λ_ ~655 A W^−1^ when operated at V_g_ = −80 V. The multilayered GeS photodetector demonstrated a high external quantum efficiency (EQE ~4.0 × 10^4^%) and specific detectivity (D* ~2.35 × 10^13^ Jones), making it highly competitive with advanced commercial Si- and InGaAs-based photodiodes, which typically exhibit detectivities in the range of 10^12^–10^13^ Jones. Additionally, the device exhibited long-term photoswitching stability over extended operation (>1 h), making 2D GeS a strong candidate for photo-assisted volatile organic compound (VOC) detection.

A recent study investigated the potential of GeS-based field-effect transistors (FETs) for volatile organic compound (VOC) sensing, demonstrating a novel mechanism based on photoactivated charge transfer interactions (Figure 18) [120]. The study examined the electrical response of GeS FETs under dark and UV light conditions, revealing a shift in semiconducting behavior when exposed to VOC species. As shown in Figure 18b, the p-type behavior of GeS was maintained under UV illumination, while in the presence of ethanol and dark conditions, the material exhibited a transition to n-type behavior (Figure 18d). This shift indicates a significant influence of VOC adsorption on carrier dynamics. The study further demonstrated that the interaction between physisorbed VOC molecules and the GeS surface (Figure 18c) could modulate carrier density, enabling differentiation between ethanol, acetone, and 2-propanol based on their distinct electrical responses. This characteristic response highlights the potential of 2D GeS for highly selective and miniaturized VOC sensors, with promising applications in health monitoring, environmental safety, and industrial process control.

A theoretical study proposed a two-dimensional (2D) germanium sulfide nanosheet (GSNS) as a high-performance anode material for AM-ion (AM = Li, Na, and K) batteries [113]. While experimental validation is still required, simulations indicate strong interaction between AM atoms and GSNS, preventing clustering and ensuring high cycling stability. The interaction between AM atoms and GSNS was found to be sufficiently strong to prevent clustering, which commonly occurs in other 2D materials. The calculated low-energy barriers for AM atom diffusion on GSNS, 0.236 eV (Li), 0.090 eV (Na), and 0.050 eV (K), suggest excellent charge/discharge rates. Additionally, the AM/GSNS system is predicted to achieve a high AM storage capacity of up to 512 mAh g^−1^ (for Na), making it a promising candidate for next-generation battery technologies.

A heterostructure composed of two-dimensional GeS and zero-dimensional gold nanoparticles (Au NPs) was used to develop an Internet-of-Things-enabled humidity sensor for human respiration monitoring. The incorporation of Au NPs significantly enhanced charge transfer efficiency, improving sensitivity and response time. The as-prepared heterostructure exhibited a maximum sensor response of 7102% at 90% relative humidity (RH), significantly outperforming pristine GeS sensors, which displayed a response of 2700% at the same RH level. Moreover, the sensor demonstrated ultrafast response and recovery times of 0.69 s and 0.73 s, respectively, making it highly suitable for real-time monitoring applications [121].

GeS has also been identified as a ferroelectric semiconductor with a large spontaneous polarization of 484 pC, which is notably higher than that of many conventional 2D ferroelectric materials, making it a strong candidate for next-generation memory devices. Few-layered GeS exhibits remarkable in-plane ferroelectric hysteresis, which is absent in bulk GeS due to inversion symmetry. A ferroelectric field-effect transistor (FeFET) based on p-type GeS demonstrated a characteristic double-hysteresis loop when the applied electric field was reversed. After poling, the few-layered GeS generated an open-circuit photovoltage of 30 mV, which could be leveraged for energy-harvesting applications and optoelectronic circuits requiring tunable voltage outputs. The photovoltage and photocurrent were switchable in response to the flipping of the ferroelectric polarization, with the direction of the photocurrent opposing the polarization vector. These findings highlight the potential of GeS-based FeFETs for next-generation nonvolatile memory and optoelectronic devices [122].

### 6.2. GeSe

GeSe-based photocathodes have demonstrated a photocurrent density of approximately −14 mA cm^−2^ and excellent long-term stability of ~60 h. However, this performance is still lower than leading photocathodes such as MoS_2_ and BiVO_4_, which have achieved photocurrent densities exceeding −20 mA cm^−2^, indicating the need for further material optimization. However, compared with state-of-the-art inorganic-based photocathodes, GeSe still requires further improvements in photocurrent density, onset potential, solar-to-hydrogen (STH) efficiency, and long-term operational stability. Strategies to enhance these parameters include suppressing intrinsic and interfacial defects; promoting large-grain growth with preferred orientation; optimizing buffer layers, hole transport layers (HTLs), and electron transport layers (ETLs); and ensuring favorable band alignments at each interface. While GeSe-based TFSCs have achieved a record certified PCE of 5.2% using the RTS method in an ITO/CdS/Sb_2_Se_3_/GeSe/Au superstrate structure, research on its PEC water-splitting application remains limited, with the highest reported STH efficiency at 3.17% [123].

Kushnir et al. [124] explored the theoretical prediction and experimental demonstration of a surface shift current response in GeSe monolayers and bulk crystals, where inversion symmetry is broken at the surface. Shift current refers to the coherent movement of electron density triggered by above-bandgap photoexcitation, which is typically forbidden in bulk GeSe due to inversion symmetry. Using terahertz (THz) emission spectroscopy, the study demonstrated that ultrafast photoexcitation at 400 nm (3.10 eV) and 800 nm (1.55 eV) induces a shift current in the surface layer of bulk GeSe, where inversion symmetry is broken. The emitted THz pulses were nearly single cycle and confirmed the presence of a spontaneous polarization-driven shift current. The direction of this surface shift current depends on the sample’s orientation and remains unaffected by the linear polarization of the excitation light. However, strong absorption by low-frequency infrared-active phonons in bulk GeSe limits the bandwidth and amplitude of the emitted THz pulses, restricting its effectiveness in high-frequency terahertz applications. The study suggests that reducing GeSe thickness to a monolayer or a few layers could significantly enhance broadband THz emission, making it a strong candidate for nonlinear photonic devices and next-generation THz sources. Reducing the thickness of GeSe could help mitigate this issue and enhance signal coherence. It is predicted that reducing the thickness of GeSe to a monolayer or a few layers could enhance terahertz emission efficiency, making GeSe a promising material for shift current photovoltaics, nonlinear photonic devices, and terahertz sources.

GeSe-based devices have also shown promising figures of merit for efficient photodetection, with performance metrics that are competitive with commercial infrared photodetectors. A Schottky barrier diode (SBD) photodetector using asymmetric Pd/Au and Cr/Au metal contacts on a p-type GeSe channel was fabricated as shown in Figure 19. The Schottky barrier diode structure enables efficient carrier separation, contributing to its high sensitivity and fast response times. Specifically, Schottky barrier diodes exhibit high sensitivity to near-infrared (NIR) light irradiation at zero bias, making them suitable for applications such as fire detection and night vision. Key performance metrics for illumination at ~850 nm include a responsivity of 280 mA/W, detectivity of 4.1 × 10^9^ Jones, a normalized photocurrent-to-dark current ratio (NPDR) of 3 × 10^7^ W^−1^, a noise-equivalent power (NEP) of 9.1 × 10^−12^ W Hz^−1^/^2^, and a response time of 69 ms. These parameters demonstrate their capability for highly sensitive, low-power photodetection, as well as highly responsive, suitable for emergency applications such as fire detection and night vision [125].

### 6.3. GeTe

Two-dimensional GeTe monolayers can be theoretically exfoliated from the bulk phase, requiring a relatively low cleavage energy of approximately 0.63 J/m^2^ [59]. GeTe monolayers exhibit semiconducting behavior with a sizable band gap of 2.35 eV and favorable band edge positions for photocatalytic water splitting, aligning well with the redox potentials of water, which enhances their efficiency in hydrogen evolution reactions. Notably, GeTe monolayers also display high electron and hole mobilities along with strong optical absorption in the visible spectrum, making them highly promising candidates for photocatalysis [59].

The nonlinear optical properties of GeTe have been investigated using Z-scan measurements at 800 nm and 1550 nm [126]. GeTe exhibits strong saturable absorption, with modulation depths of 16.5% at 800 nm and 3.7% at 1550 nm. Mode-locking experiments demonstrated ultrashort pulse durations of 680 fs centered at 1562 nm. Additionally, stable Q-switched pulse trains were achieved, featuring a maximum output power of 43.2 mW, a maximum pulse energy of 315 nJ, a shortest pulse duration of 0.978 μs, and a tunable repetition rate ranging from 74.5 kHz to 137 kHz. Compared with other saturable absorbers, GeTe offers competitive nonlinear optical properties, making it a promising material for ultrafast laser applications. These results highlight the potential of GeTe nanosheets as efficient saturable absorbers for ultrafast photonics applications.

The photocatalytic hydrogen production potential of 2D GeTe was experimentally investigated by Zhang et al. [60]. Their study demonstrated that different layered GeTe samples act as indirect-gap semiconductors, with band gap widening occurring after oxidation. All tested samples exhibited appropriate band positions for photocatalytic water splitting under mild conditions, achieving maximum hydrogen evolution rates of 1.13 mmol g^−1^ h^−1^ (for Ar-GeTe) and 0.54 mmol g^−1^ h^−1^ (for O-GeTe) as shown in Figure 20. Density functional theory (DFT) calculations further indicated that oxygen atoms readily interact with Ge to form a more stable oxidized structure, which slightly reduces the material’s photocatalytic efficiency but enhances its long-term stability under ambient conditions. The ability to operate under light-limited and oxygen-deficient environments suggests GeTe’s potential for space-based energy applications.

A GeTe-based photodetector was fabricated by sputtering a nanofilm of SnTe onto a pre-masked n-Ge substrate. J-V measurements of the SnTe/n-Ge heterostructure revealed diode and photovoltaic characteristics across a broad spectral range (400–2050 nm). Under 850 nm NIR illumination with an optical power density of 13.81 mW cm^−2^, the photodetector exhibited an open-circuit voltage of 0.05 V. Additionally, it achieved a high responsivity of 617.34 mA W^−1^ (at −0.5 V bias) and a detectivity of 2.33 × 10^11^ cmHz^1^/^2^W^−1^ (at zero bias). These results demonstrate the potential of SnTe nanofilm/n-Ge heterostructures as low-power, broadband photodetectors with simple device configurations, rivaling existing commercial photodetectors in responsivity and detectivity while maintaining cost-effective fabrication [89].

### 6.4. SnS

Two-dimensional SnS exhibits a broad spectrum of potential applications. It has demonstrated strong catalytic activity, particularly in hydrogen evolution reactions, where it shows an impressive hydrogen generation rate [127]. Enhanced photocatalytic performance has been observed in Sn/SnS_2_ heterostructures, with the SnS/SnS_2_ heterojunction exhibiting a photocurrent density up to 10 times higher than that of pure SnS_2_ and 6 times higher than that of pure SnS, highlighting its improved charge separation efficiency and visible-light-assisted electrochemical water-splitting capabilities [128]. Additionally, SnS’s diverse morphologies provide flexibility for different catalytic applications. SnS nanocrystals have demonstrated remarkable photocurrent density of 7.6 mA/cm^2^ under 100 mW/cm^2^ illumination, approximately 10 times higher than that of bulk SnS, with an incident photon-to-current conversion efficiency (IPCE) of 9.3% at 420 nm compared with 0.78% for bulk SnS [129]. ZnS/SnS/A-FA nanorods have also shown efficient degradation of Congo red dye within 150 min and significant antibacterial activity against both Gram-positive and Gram-negative bacteria, highlighting their potential for wastewater treatment and antimicrobial applications [130].

In photonics, SnS possesses a superior nonlinear optical absorption coefficient compared with black phosphorus, measured to be an order of magnitude higher, enhancing its potential for ultrafast laser generation by enabling efficient pulse compression and high-speed optical modulation. SnS exhibits a tunable optical modulation depth with a maximum value of 36.4%, making it a promising material for mode-locked and Q-switched fiber lasers. Additionally, its response time, characterized by femtosecond-resolved transient absorption spectra, indicates potential operation speeds in the gigahertz to terahertz range [131]. Its absorption peaks in the near-infrared range also present potential advantages for telecommunication applications [131].

Thin SnS layers produced through PVD have demonstrated exceptional photoresponsivity, reaching 365 A/W under 808 nm light illumination, with an external quantum efficiency (EQE) of 5.70 × 10^4^%. The performance is further enhanced by Au nanoparticle decoration, increasing the photoresponsivity to 635 A/W and the EQE to 9.92 × 10^4^% [132]. Exfoliated SnS nanosheets enable the fabrication of tunable bandgap photodetectors with high responsiveness, exhibiting a broad absorption range from 380 nm to 1000 nm. These devices achieve a high photocurrent density of 1590 nA/cm^2^ and a photoresponsivity of 59.8 μA/W under neutral Na_2_SO_4_ aqueous solutions, maintaining strong stability in acidic, neutral, and basic electrolytes for over one month without degradation [25]. Moreover, SnS-based devices have exhibited robustness in demanding environments, contributing to their suitability for long-term applications [25].

SnS has also proven to be an effective chemical sensor, particularly in NO_2_ detection. The integration of 2D SnS nanoflakes into a resistive transducing platform has demonstrated an excellent response magnitude of ~68% at 3750 ppb NO_2_ concentration, with a remarkably low limit of detection (LOD) of 17 ppb at an operating temperature of 60 °C as seen in Figure 21. These results significantly surpass those of previously reported 2D p-type semiconductor-based NO_2_ sensors, making SnS a highly promising material for chemical sensing applications [133]. SnS-based humidity sensors have further been developed for biomedical applications, offering an exceptionally high response of 2,491,000% across a wide relative humidity (RH) range from 3% to 99%. The sensors exhibit rapid response and recovery times of 6 s and 4 s, respectively. Furthermore, flexible SnS nanoflake-based humidity sensors integrated onto polyimide substrates demonstrate stable performance under bending conditions, making them suitable for wearable and non-contact physiological monitoring, such as respiration pattern detection and fingertip movement sensing [134].

In biomedical applications, SnS has been explored for its biocompatibility, large surface area, and strong near-infrared absorption. PEGylated SnS nanosheets exhibit significantly higher extinction coefficients and photothermal conversion efficiencies compared with bulk SnS, making them highly effective for photothermal therapy. Upon NIR light irradiation, these nanosheets efficiently induce cancer cell death in vitro, demonstrating their potential as photothermal agents for cancer treatment [135]. These properties make it a viable material for photothermal therapy and combined photothermal–chemotherapeutic treatments, showing promise in cancer therapy [136].

SnS’s layered structure enables efficient ion intercalation, enhancing charge storage capacity and cycling stability, making it a strong candidate for energy storage applications. In lithium-ion batteries, self-supported 2D SnS nanosheet electrodes grown via a non-catalytic vapor transport method have demonstrated a high discharge capacity exceeding 560 mAh/g after 50 cycles, with a rate capability of 511 mAh/g at 10C, surpassing commercial carbon-based electrodes [137]. For sodium-ion batteries, SnS/CNT@C composites have exhibited outstanding performance, retaining a specific capacity of 494.9 mAh/g after 400 cycles at 1.0 A/g and achieving a rate capacity of 431 mAh/g at 5.0 A/g [138]. In potassium-ion batteries, SnS@C/rGO composites have delivered a high capacity of 565 mAh/g at 100 mA/g, with stable cycling performance optimized by adjusting the voltage window to prevent deep alloying reactions, further enhancing electrochemical performance [139].

### 6.5. SnSe

In a study by Wang et al. [140], the nonlinear saturable absorption characteristics of 2D SnSe were investigated in the near- and mid-infrared spectral regions. The modulation depths at 1.5 µm and 2.0 µm were found to be 14.4% and 37%, respectively. By integrating SnSe onto a microfiber-based device, stable pulse generation was achieved in ultrashort pulse fiber lasers, producing pulse durations of 542 fs at 1.5 µm and 2.12 ps at 2.0 µm as shown in Figure 22. Furthermore, SnSe-decorated devices enabled harmonic mode-locking bound pulses at 1.5 µm fiber lasers, with a tunable frequency range of 29.7 to 108.9 MHz, demonstrating the potential for high-speed optical communication and ultrafast laser applications. At 2.0 µm, the fiber laser generated dual-wavelength pulses at 1897.3 nm and 1910.5 nm, with spectral bandwidths of 2.13 nm and 0.15 nm, respectively.

CVD-grown SnSe crystals exhibit excellent polarization detection capabilities, with an anisotropic photocurrent ratio of 2.31 at 1064 nm due to the van der Waals superposition of covalently bonded atomic layers. Additionally, SnSe-based photodetectors demonstrate a high responsivity of 9.27 A/W, a detectivity of 4.08 × 10^10^ Jones, and response times in the nanosecond range [141]. Furthermore, PVT-synthesized SnSe has shown memristive behavior in two-terminal lateral devices, with a low threshold voltage of just 3 V and an operating current of 10^−4^ A, offering lower power consumption compared with similar devices [13].

SnSe has also emerged as a highly promising thermoelectric material, exhibiting exceptional performance in both n-type and p-type crystalline forms. Recent advancements have demonstrated that incorporating magnetic nanoprecipitates, such as Gd_2_Se_3_, into SnSe nanoplates enhances its thermoelectric properties. This approach has resulted in a thermoelectric power factor of 6.7 µW/cmK^2^ at 868 K and a reduced lattice thermal conductivity of 0.41 W/mK, leading to a peak ZT value of ~1 at 868 K in spark plasma sintered (SPS) SnSe nanoplates [142]. Its anisotropic crystal structure and unique chemical bonding facilitate efficient electrical transport along the in-plane direction, while strong anharmonicity suppresses thermal transport, particularly out of plane. Tuning the carrier concentration activates multiple electronic bands, enhancing the Seebeck coefficient and power factor. Furthermore, phase transitions and overlapping electron orbitals in the conduction bands improve electron flow in the out-of-plane direction, resulting in more isotropic three-dimensional electron transport, an effect not observed in hole transport.

In solar cells, SnSe—whether in thin-film or quantum dot form—exhibits a high absorption coefficient, an optimal bandgap, superior carrier mobility, and strong catalytic activity. However, despite continued advancements, SnSe-based thin-film solar cells have achieved efficiencies of only 2.51%, significantly below the theoretical maximum of ~27.7%. Meanwhile, SnSe has also been explored as a counter electrode material in dye-sensitized solar cells (DSSCs), where it has demonstrated efficiencies of 5.8% and 4.9%, showing promise for alternative solar energy applications [143].

SnSe has also demonstrated potential as an electrode material in rechargeable batteries, including lithium-ion, sodium-ion, potassium-ion, and aluminum-ion batteries. Its layered crystal structure provides a large surface area and weak van der Waals interlayer interactions, enabling efficient charge transfer and high reversible capacity. For lithium-ion batteries, SnSe has shown specific capacities reaching 780 mAh/g at low current densities, with cycling stability improving significantly when SnSe is combined with carbon-based composites. In sodium-ion batteries, SnSe-based anodes have achieved a reversible capacity of 510 mAh/g after 200 cycles at 0.1 A/g, demonstrating promising electrochemical performance. Meanwhile, for potassium-ion batteries, SnSe has exhibited a stable reversible capacity of ~400 mAh/g over 100 cycles, benefiting from structural modifications to accommodate large K^+^ ions. To further enhance electrochemical performance, strategies such as expanding lattice spacing to create more reactive sites, designing hollow or porous structures for improved cycling stability, and incorporating bimetal alloys to enhance charge transfer kinetics are being explored [143].

### 6.6. SnTe

CVD growth of ultrathin SnTe nanoplates (~3.6 nm thick) on mica substrates has been utilized for near-infrared (NIR) photodetectors. These SnTe nanoplate photodetectors exhibit high flexibility, responsivity, and detectivity to NIR light at 980 nm, with performance metrics including a responsivity of 698 mA/W, an external quantum efficiency (EQE) of 88.5%, and a detectivity of 3.89 × 10^8^ Jones as shown in Figure 23. These values indicate strong light absorption and charge carrier generation, making them highly competitive for advanced photodetection applications. On a SiO_2_ substrate, performance was further enhanced, achieving a responsivity of 723 mA/W and a detectivity of 5.3 × 10^8^ Jones, making these photodetectors competitive with or superior to those based on other 2D nanomaterials [114].

A photovoltaic detector was developed using a SnTe/n-Ge heterostructure via a one-step magnetron sputtering technique. This heterostructure demonstrated excellent diode rectification characteristics within a narrow bias range (−0.5 V to 0.5 V) and exhibited broadband photovoltaic properties from visible light to near-infrared (400–2050 nm). Under 850 nm illumination, the photodetector achieved a responsivity of 617.34 mA/W (at −0.5 V bias) and a detectivity of 2.33 × 10^11^ cmHz^1^/^2^ W^−1^ at zero bias [89].

SnTe multilayers have been successfully employed in developing all-in-one spin transistors that utilize both the persistent spin helix (PSH) and spin Hall effect (SHE). The combination of these effects allows for long spin coherence lengths and efficient charge-to-spin conversion, which enhances the performance of spintronic devices. First-principles calculations suggest that these devices exhibit a natural thickness limit of approximately three monolayers (~20 Å). Additionally, ultrathin SnTe films demonstrate a moderate enhancement of intrinsic spin Hall conductivity, further improving their spintronic potential. The emergence of PSH is strictly dependent on maintaining a polar stacking order that preserves Pmn2_1_ symmetry as perturbations such as substrate interactions may disrupt spin protection and reduce spin lifetime [144]. The combination of these effects allows for long spin coherence lengths and efficient charge-to-spin conversion, which enhances the performance of spintronic devices. These devices benefit from the long spin lifetime enabled by PSH and the efficient spin/charge interconversion without the need for ferromagnetic electrodes. While challenges such as a three-layer thickness limit exist, multilayers maintain polar stacking order, and substrate effects must be considered. To maximize spatial symmetry preservation, these systems are best constructed in a sandwich configuration [144].

SnTe nanosheets have also demonstrated potential in biomedical applications, offering NIR optical activity and good biocompatibility. NIR light penetration allows deeper tissue targeting, making these nanosheets suitable for advanced diagnostics and photothermal cancer treatment. Zhang et al. [145] developed a SnTe nanosheet-based diagnostic imaging and photothermal therapy agent using ball-milling and liquid exfoliation methods. The prepared SnTe@MnO_2_-SP NSs exhibited a notably high photothermal conversion efficiency of 38.2% in the NIR I window and 43.9% in the NIR II window, demonstrating their strong potential for deep-tissue cancer treatment. Additionally, these nanosheets provided high-resolution fluorescence, photoacoustic, and photothermal imaging capabilities, making them effective multimodal imaging agents. Their tumor microenvironment-responsive biodegradability, coupled with TeO_3_^2^− metabolite-induced chemotherapy, further enhances their therapeutic efficacy [145]. The prepared SnTe NSs were further modified with a soybean phospholipid (SP) and MnO_2_ shell (SnTe@MnO_2_-SP nanosheets), which exhibited excellent stability in physiological environments and unique biodegradability.

The effect of layer thickness on quantum confinement in 2D SnTe single crystals has also been explored. Li et al. [146] found that quantum confinement is highly sensitive to layer thickness, with a critical size identified at three layers (approximately 2 nm). Below this threshold, strong quantum confinement effects enhance electronic properties, while above this limit, mobility and thermoelectric parameters degrade significantly. Below this threshold, band structure modifications lead to enhanced carrier mobility and electronic properties, whereas beyond this limit, quantum confinement effects diminish. Beyond this threshold, mobility and thermoelectric properties degrade significantly. The study revealed that monolayer SnTe exhibited the highest thermoelectric performance, with optimal power factor (PF) and ZT values. As thickness increased beyond three layers, the degradation in carrier lifetime and mobility resulted in reduced thermoelectric efficiency, approaching bulk-like behavior at six layers. This layer-dependent effect was attributed to changes in relaxation time, effective mass, electrical conductivity, and the Seebeck coefficient. Consequently, PF and ZT values demonstrated strong tunability with variations in layer thickness. The highest ZT values were obtained for monolayers, while thicker films exhibited reduced thermoelectric efficiency due to increased thermal conductivity and lower carrier mobility. For instance, optimized SnTe monolayers exhibited enhanced ZT values, whereas bulk-like structures suffered from higher thermal conductivity and reduced efficiency.

Table 6 summarizes the applications of 2D group IV monochalcogenides, along with their synthesis methods and performance metrics as reported in recent studies. The data include photodetection, photocatalysis, ferroelectric memory, thermoelectrics, and more, highlighting each material’s unique properties and potential for advanced device applications.

In the table above, each material shows unique strengths: GeS excels in photodetection and ferroelectric effects; GeSe excels in self-powered and ultrafast photonics; GeTe excels in photocatalysis; SnS and SnSe excel in broad-spectrum and polarized light sensing (as well as energy applications); and SnTe excels in flexible IR detection, spintronics, and even biomedical uses. These recent advances, enabled by high-quality 2D material synthesis via CVD, LPE, and ME, highlight the rapid development of group IV monochalcogenides for next-generation devices.

## 7. Perspectives and Future Directions

Despite significant theoretical advancements, experimental progress faces challenges such as monolayer synthesis, environmental instability, and discrepancies between predicted and observed properties.

### 7.1. Challenges in Synthesis and Stability

A primary challenge lies in the group IV monochalcogenides environmental instability. Monolayers of these materials exhibit strong interactions with water molecules, leading to rapid degradation compared with other 2D systems like graphene or TMDs. The polar nature of monolayers amplifies this issue as the uneven charge distribution facilitates hydrolysis. Recent studies suggest that using nonpolar solvents during synthesis and encapsulation with hydrophobic layers could mitigate degradation, but these strategies remain experimentally unproven at scale [107].

The experimental realization of high-quality monolayers remains a significant bottleneck. While bulk crystals can be exfoliated mechanically, achieving atomically thin sheets with uniform thickness is challenging due to the strong interlayer van der Waals interactions in some compositions. Chemical vapor deposition and molecular beam epitaxy have shown promise for SnSe and GeSe, but these methods often yield multilayered structures or introduce defects. For instance, SnS monolayers grown via physical vapor deposition frequently exhibit inhomogeneous strain and surface oxidation, compromising their electronic performance [106].

Scaling up production while maintaining material quality is another major challenge. Current synthesis routes, such as laser etching combined with mechanical exfoliation, are labor-intensive and unsuitable for industrial applications. Reproducibility issues arise from subtle variations in precursor ratios, temperature gradients, and substrate interactions, which disproportionately affect the anisotropic properties of these materials. Large single crystals with few-layer thickness have been synthesized for SnS and SnSe, but extending this to monolayers remains challenging [148].

### 7.2. Electronic and Optical Property Limitations

While theoretical studies predict direct bandgaps ranging from 1.5 to 2.5 eV for monolayers, ideal for optoelectronic applications, experimental measurements often show indirect gaps or smaller values due to defects and interfacial interactions. For example, monolayer SnS exhibits a predicted direct gap of 2.1 eV, but synthesized samples frequently show indirect transitions at 1.7 eV, reducing their efficiency in photovoltaic devices. Carrier mobilities, although theoretically high (∼300 cm^2^/V·s for electrons in SnSe), are limited by phonon scattering and impurity concentrations in practice [109].

The valley polarization effects in group IV monochalcogenides, important for valleytronics, are highly sensitive to layer thickness and symmetry. Monolayers exhibit spin-orbit splitting of 48–87 meV in the conduction band due to their non-centrosymmetric structure. However, even-numbered layers regain inversion symmetry, nullifying this splitting and complicating device design. Maintaining valley polarization at room temperature remains unachieved, as thermal excitations erase the valley contrast [106].

### 7.3. Perspectives on Emerging Applications

The anisotropic phonon and electron transport in group IV monochalcogenides position them as leading candidates for thermoelectric materials. Their puckered structure decouples thermal and electrical conductivity along orthogonal directions, a phenomenon first observed in phosphorene. SnSe monolayers theoretically exhibit a thermoelectric figure of merit exceeding 2.5 at 700 K, surpassing traditional materials such as Bi_2_Te_3_ [149]. Experimental validation of these values requires advances in reducing interfacial thermal resistance in thin-film devices.

The non-centrosymmetric structure of monolayers enables robust piezoelectric responses, with coefficients reaching 250–350 pm/V, significantly higher than MoS_2_ (∼15 pm/V) [150]. This property, combined with inherent ferroelectricity from switchable out-of-plane dipoles, creates opportunities for self-powered sensors and low-energy memory devices. However, achieving reversible polarization switching without fatigue remains a challenge.

The strong optical absorption (∼10^5^ cm^−1^) in the visible to near-infrared spectrum makes these materials suitable for photodetectors and solar cells. SnS-based photodetectors have demonstrated responsivities of 22 A/W, but response times (>1 ms) lag behind those of commercial Si-based devices [109]. Integrating monolayers with transparent conductors like graphene could enhance performance while addressing contact resistance issues.

### 7.4. Future Research Directions

Developing substrate-free growth techniques and optimizing precursor delivery in CVD/MBE systems are essential for monolayer reproducibility. In situ characterization tools, such as scanning tunneling microscopy combined with Raman spectroscopy, could enable real-time monitoring of layer formation and defect dynamics [106]. Additionally, exploring liquid-phase exfoliation with ionic liquids may improve yield and scalability.

Constructing van der Waals heterostructures with complementary 2D materials (e.g., graphene and h-BN) could mitigate stability issues and enhance device functionality [151]. For instance, encapsulating SnSe between h-BN layers may preserve its piezoelectric properties while shielding it from moisture [152]. Computational screening of alloy combinations (e.g., SnS_x_Se_1−x_) could identify compositions with tailored bandgaps and improved thermal stability.

The high ductility of group IV monochalcogenides allows significant bandgap tuning (∼200 meV per 1% strain). Integrating monolayers onto flexible substrates with pre-strained patterns could enable strain-engineered optoelectronic devices. Similarly, applying out-of-plane electric fields could induce reversible structural transitions, offering a path for non-volatile memory applications [150].

Transitioning from material characterization to functional devices requires addressing contact resistance, doping control, and interfacial charge transfer. Developing selective area doping techniques using plasma treatments or molecular adsorbates could create p-n junctions within monolayers. Standardizing performance metrics across laboratories will facilitate benchmarking against existing technologies.

## 8. Conclusions

This review has provided an in-depth analysis of the synthesis, structural properties, and applications of group IV monochalcogenides, a unique class of chalcogenide materials that exhibit promising potential in nanoelectronics, optoelectronics, and energy storage. These materials possess distinctive electrical, optical, and mechanical properties, including high carrier mobility for electronic applications, strong light absorption for optoelectronics, and excellent mechanical flexibility for flexible devices, making them highly attractive for various technological advancements.

We began by discussing the structural characteristics of monochalcogenides, highlighting their diverse crystallographic configurations and phase transitions. The review then explored the most commonly used synthesis techniques, such as chemical vapor deposition and physical vapor deposition, which allow for scalable and high-quality production of 2D monochalcogenides.

The electronic properties of these materials were examined, emphasizing their semiconducting behavior; tunable bandgap engineering; and the influence of dimensionality and defects on charge transport, carrier recombination, and overall device efficiency. In optoelectronics, monochalcogenides have demonstrated great potential in applications such as photodetectors, light-emitting diodes (LEDs), and other photonic devices. Additionally, their integration into energy storage technologies, including lithium-ion batteries and supercapacitors, showcases their versatility in sustainable energy solutions.

While defects in materials are often associated with undesirable effects, controlled defect engineering in group IV monochalcogenides can, in some cases, enhance their properties rather than degrade them. Atomic vacancies and other structural imperfections have been shown to modulate electronic structures, improve catalytic activity, and optimize thermoelectric performance. As a result, even with certain intrinsic defects, these materials can still exhibit high performance in optoelectronic, energy, and sensing applications.

Despite these advancements, several challenges remain, including refining synthesis methods for large-scale industrial applications, addressing degradation mechanisms that affect long-term material stability, and improving device integration strategies to maximize efficiency and performance. Future research should focus on overcoming these challenges through advanced fabrication techniques such as molecular beam epitaxy and atomic layer deposition, strategic defect engineering to enhance charge carrier dynamics, and the exploration of heterostructures for multifunctional device applications, further expanding the potential of group IV monochalcogenides in emerging technologies.

## Figures and Tables

**Figure 1 materials-18-01530-f001:**
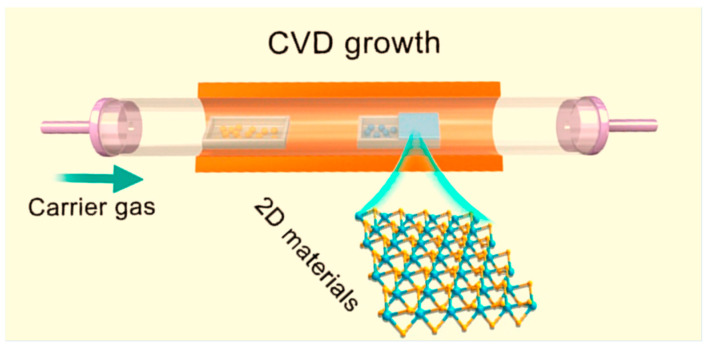
Schematic of the CVD growth of 2D group IV MCs. Reproduced with permission from [35].

**Figure 2 materials-18-01530-f002:**
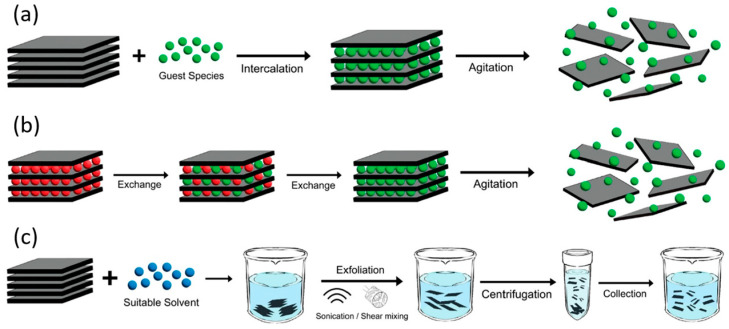
Schematic description of the main liquid-phase exfoliation mechanisms: (**a**) ion intercalation, (**b**) ion exchange, and (**c**) sonication/shearing-assisted exfoliation. Reproduced with permission from [48].

**Figure 4 materials-18-01530-f004:**
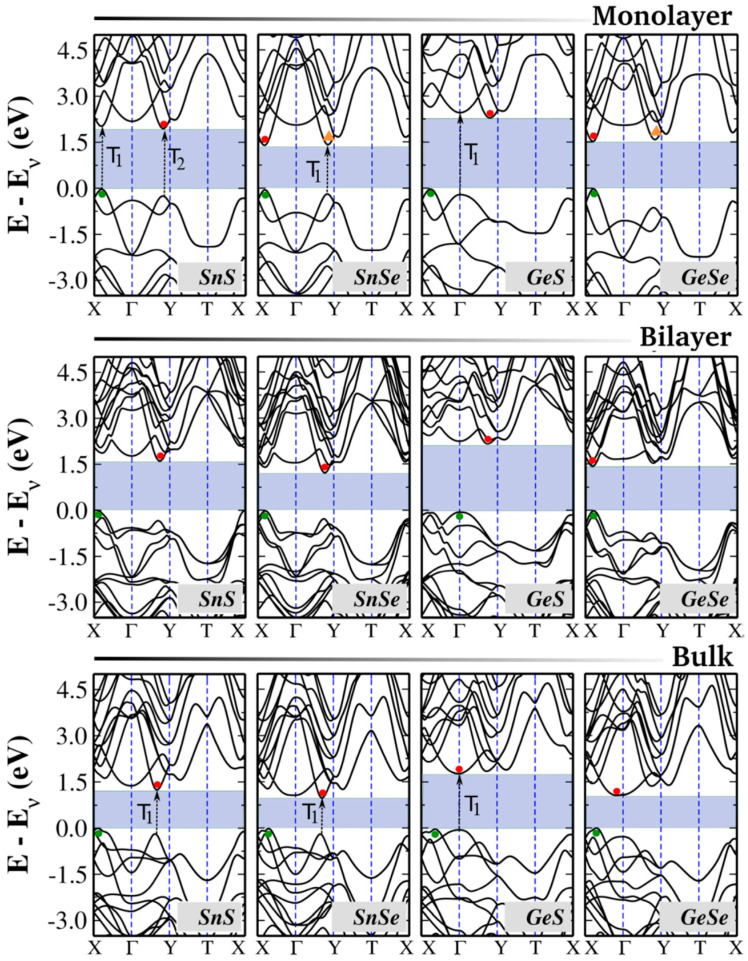
Electronic band structures of monolayer, bilayer, and bulk group IV monochalcogenides. The valence band maximum (VBM) and conduction band minimum (CBM) are marked with solid circles. Dashed black arrows indicate possible direct transitions (T1 and T2) to points very close in energy to the VBM and CBM. The triangles represent the CBM positions when spin–orbit coupling (SOC) effects are included. Reproduced with permission from [80].

**Figure 5 materials-18-01530-f005:**
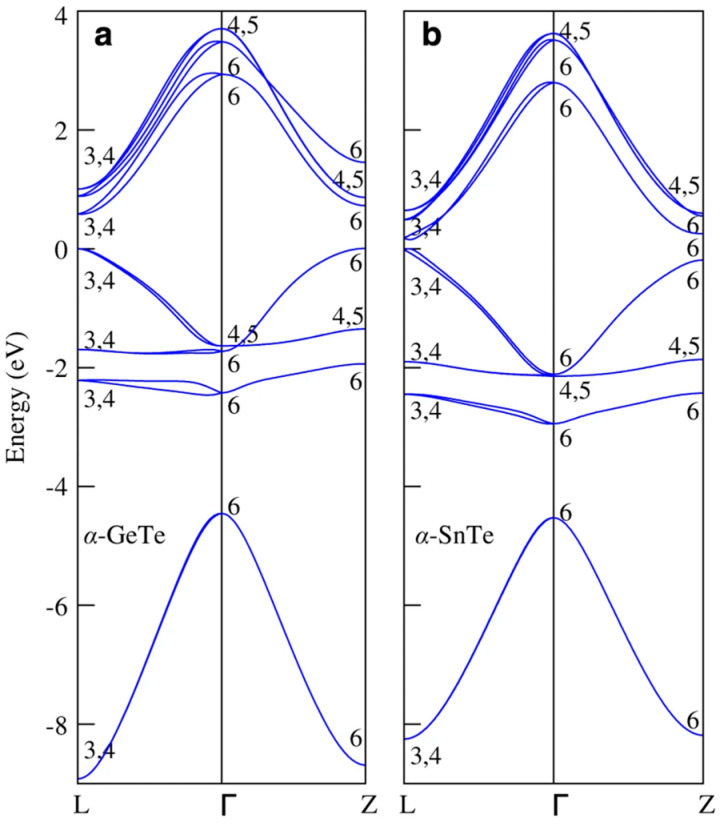
Calculated energy band structure of (**a**) GeTe and (**b**) SnTe in rhombohedral structure. Reproduced with permission from [101].

**Figure 6 materials-18-01530-f006:**
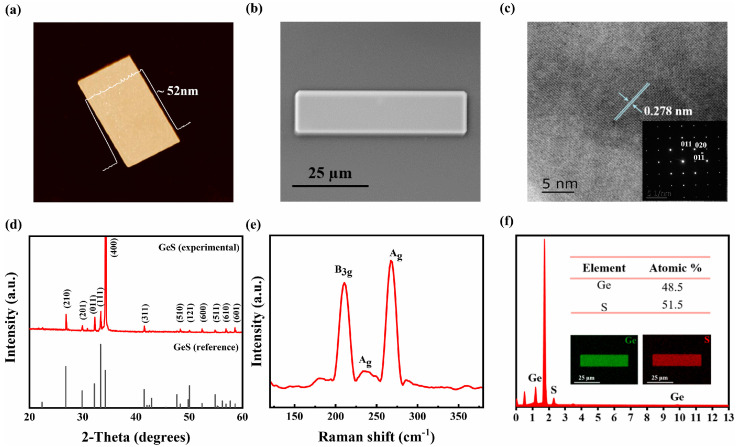
Structural and compositional characterization of GeS: (**a**) AFM image showing the flake thickness, (**b**) SEM image displaying surface morphology, (**c**) HRTEM image with SAED inset confirming crystallinity, (**d**) XRD comparison with standard orthorhombic GeS, (**e**) Raman spectra with characteristic vibrational modes, and (**f**) EDS analysis showing elemental composition and distribution. Reproduced with permission from [105].

**Figure 7 materials-18-01530-f007:**
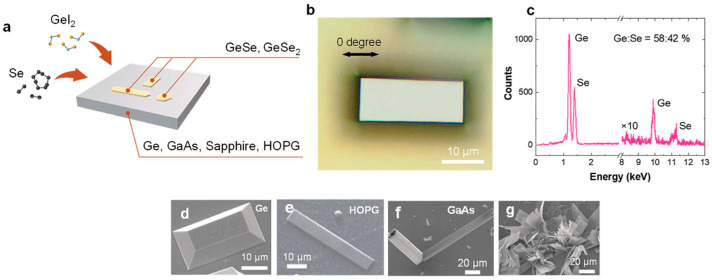
GeSe characterization: (**a**) Schematic description of GeSe formation process on various substrates. (**b**) Optical image of a GeSe flake. (**c**) EDX analysis of GeSe. (**d**–**g**) SEM images of Ge, HOPG, GaAs, and GeSe flakes. Reproduced with permission from [37].

**Figure 8 materials-18-01530-f008:**
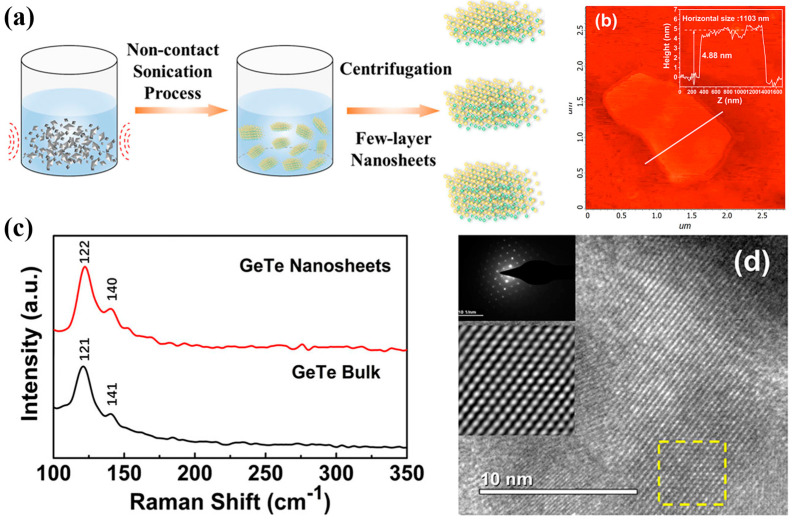
Synthesis and structural characterization of GeTe nanosheets: (**a**) Schematic diagram of the liquid-phase exfoliation process under ultrasound. (**b**) AFM morphology images of GeTe nanosheets, with an inset showing the height profile along a white line measuring 1103 nm in lateral size and 4.85 nm in thickness. (**c**) Raman spectra comparing bulk and exfoliated GeTe nanosheets. (**d**) HRTEM image of GeTe nanosheets, with insets displaying the corresponding SAED patterns (**top**) and FFT-filtered images (**bottom**) of the yellow-dotted square region. Reproduced with permission from [112].

**Figure 9 materials-18-01530-f009:**
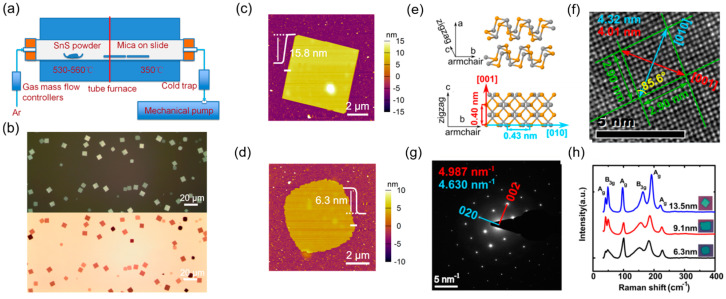
SnS characterization techniques and structural analysis: (**a**) Schematic of the PVT growth system used for growing 2D SnS nanoplates on mica substrates. (**b**) Optical microscopy images of SnS nanoplates on mica, shown in both reflection (**top**) and transmission (**bottom**) modes. (**c**,**d**) AFM images of SnS nanoplates with thicknesses of 15.8 nm and 6.3 nm, respectively, transferred onto SiO_2_ substrates. (**e**) Atomic model of bulk SnS crystal structure viewed in 3D perspective (**top**) and along the interlayer direction (**bottom**). (**f**) HRTEM image of an as-grown 2D SnS nanoplate, showing lattice fringes along the [011], [010], and [001] directions, matching the bulk SnS structure. (**g**) SAED pattern corresponding to (**f**), showing interplanar distances of 4.987 nm (cyan line) and 4.630 nm (red line). (**h**) Raman spectra of SnS nanoplates with thicknesses of 6.3 nm, 9.1 nm, and 13.5 nm, highlighting characteristic A_g_ and B_3g_ vibrational modes with improved sharpness for thicker samples. Reproduced with permission from [64].

**Figure 10 materials-18-01530-f010:**
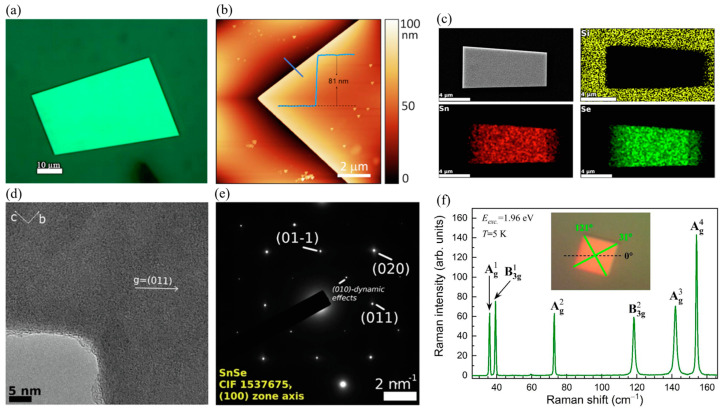
Structural and optoelectronic characterization of SnSe nanoflakes: (**a**) Optical image of a synthesized SnSe nanoflake on a Si/SiO_2_ (300 nm) substrate. (**b**) AFM image showing a thickness below 100 nm. (**c**) SEM image with EDX elemental mapping of a SnSe nanoflake. (**d**) HRTEM image of a SnSe nanoflake, with the inset showing the SAED pattern. (**e**) SAED pattern observed along the [100] zone axis, confirming the single-crystalline orthorhombic nature. (**f**) Raman spectrum measured at low temperature (T = 5 K), revealing phonon modes and structural anisotropy. Reproduced with permission from [13].

**Figure 11 materials-18-01530-f011:**
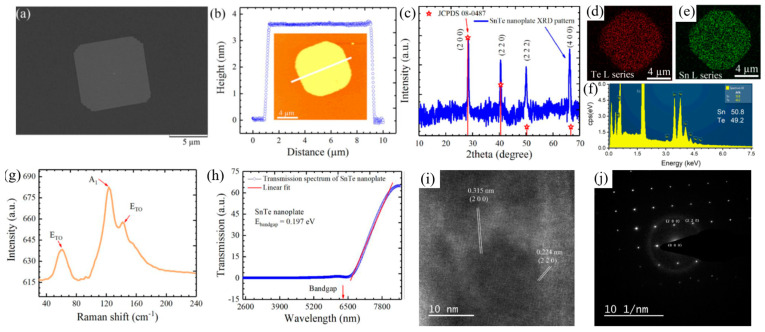
Comprehensive characterization of SnTe nanoplates: (**a**) SEM image of a representative SnTe nanoplate on a Si substrate; (**b**) AFM thickness profile showing 3.6 nm; (**c**) XRD pattern compared to JCPDS data; (**d**–**f**) EDS mapping confirming uniform Sn and Te distribution; (**g**) Raman spectrum showing peaks at 123, 139, and 61 cm^−1^; (**h**) Transmission spectrum with a 6.8 μm cutoff wavelength corresponding to a bandgap of 0.197 eV; (**i**) HRTEM image showing interplanar spacings of (200) and (220) planes; (**j**) SAED pattern confirming the tetragonal structure. Reproduced with permission from [114].

**Figure 12 materials-18-01530-f012:**
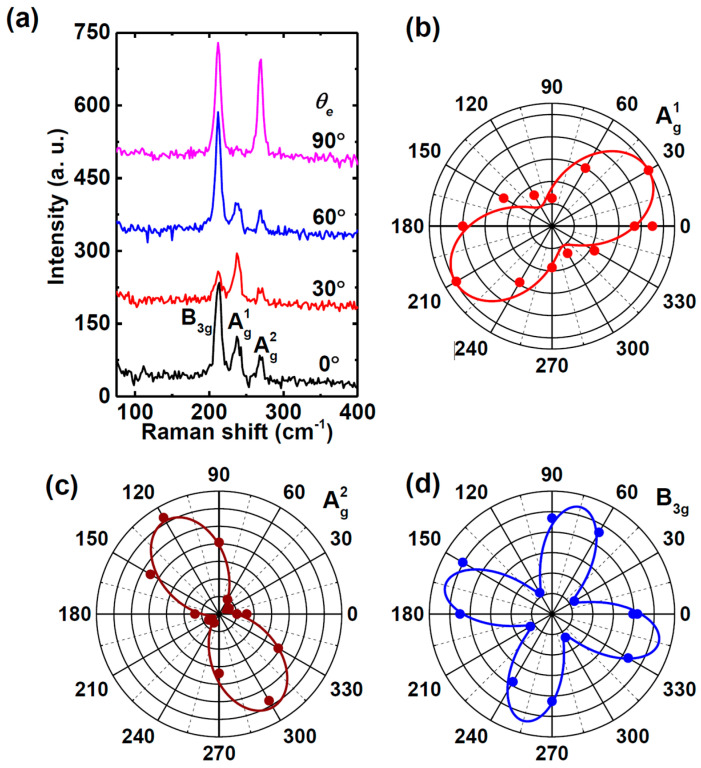
Polarized Raman spectroscopy of GeS nanoflakes: (**a**) Raman-scattering spectra of layered GeS at various angles in a parallel-polarization (//) configuration. The zero-angle direction is defined as a reference at the start of the experiment. (**b**–**d**) Polar plots of Raman peak intensities for A^1^_g_, A^2^_g_, and B_3g_ modes, respectively, as functions of the excitation polarization angle θe. Red curves indicate fitting results based on theoretical models. Reproduced with permission from [115].

**Figure 13 materials-18-01530-f013:**
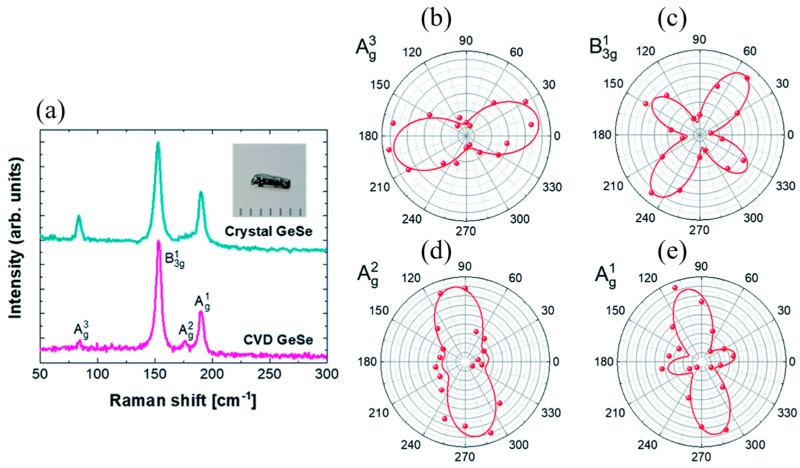
Raman and structural characterization of GeSe: (**a**) Comparison between Raman spectra of bulk and CVD-grown GeSe. (**b**–**e**) Angle-resolved Raman spectra contour plots for different Raman modes, illustrating vibrational anisotropy. Reproduced with permission from [37].

**Figure 14 materials-18-01530-f014:**
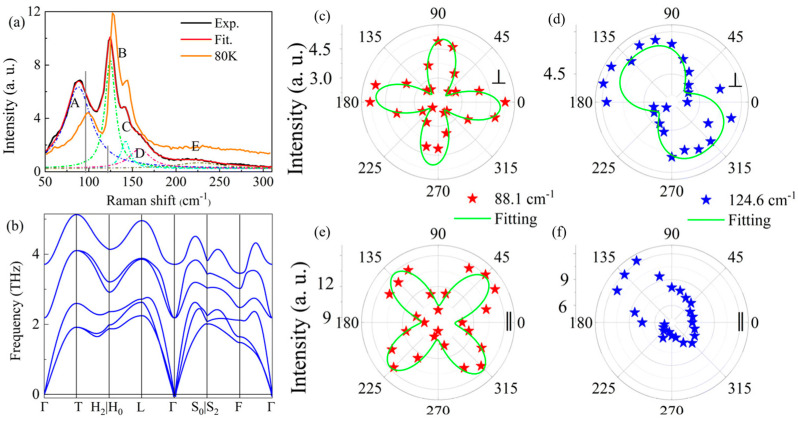
Raman characterization and polarization dependence of GeTe: (**a**) Raman spectrum of bulk GeTe at room temperature, showing experimental results (red) and calculated phonon modes (black). Fitting identified five peaks at 88.1 cm^−1^ (A), 124.6 cm^−1^ (B), 142.2 cm^−1^ (C), 158.4 cm^−1^ (D), and 225.5 cm^−1^ (E). (**b**) Calculated phonon dispersion relations of bulk GeTe. (**c**–**f**) Polar plots of peak intensity for E and A_1_ modes under parallel and cross-polarization configurations, with experimental values (stars) and fitted results (green curves). Reproduced with permission from [116].

**Figure 15 materials-18-01530-f015:**
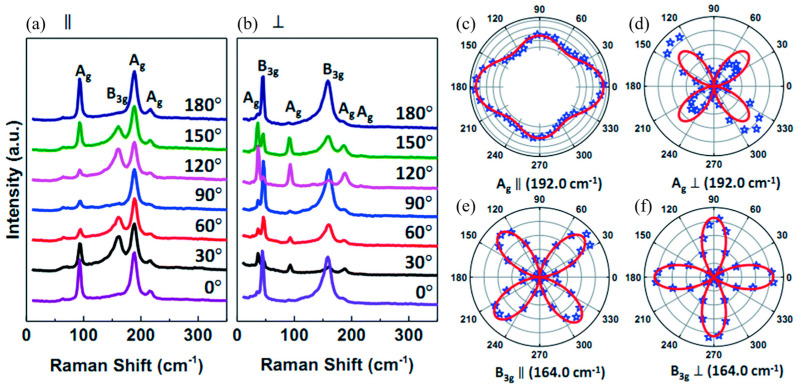
Polarization-dependent Raman response of SnS nanoflakes: (**a**) Parallel and (**b**) perpendicular polarization configurations used for Raman measurements. (**c**–**f**) Polar plots of the intensity variations of A_g_ (192 cm^−1^) and B_3g_ (164.0 cm^−1^) modes under parallel and perpendicular polarization, respectively, demonstrating the anisotropic vibrational properties of SnS with experimental values (blue stars) and fitted results (red curves). Reproduced with permission from [117].

**Figure 16 materials-18-01530-f016:**
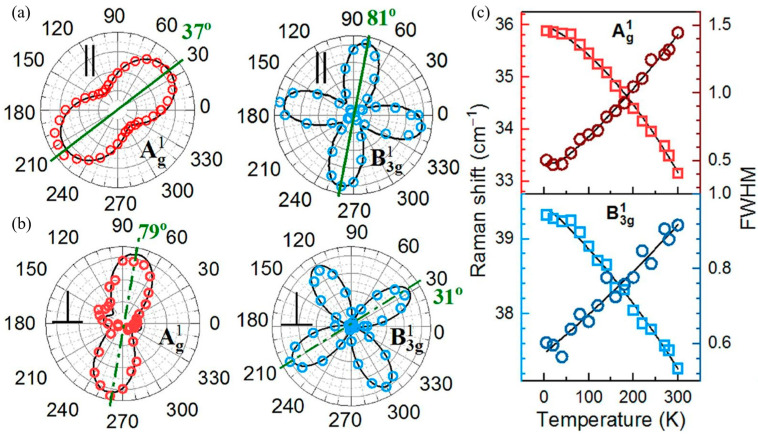
Anisotropic Raman response and temperature dependence of phonon modes in SnSe nanoflakes: (**a**) Polar plots of the integrated intensities of A^1^_g_ and B_3g_ phonon modes in the ∥ configuration at 5 K under 1.96 eV excitation, with green solid lines indicating polarization axes, symbols indicating experimental values and black curves indicating the fitted results. (**b**) Similar polar plots for A^1^_g_ and B_3g_ phonon modes in the ⊥ configuration, showing a 45° shift relative to the ∥ configuration. (**c**) Temperature dependence of phonon energies measured on SnSe nanoflakes, illustrating anharmonic effects and phonon–phonon interactions. Reproduced with permission from [13].

**Figure 17 materials-18-01530-f017:**
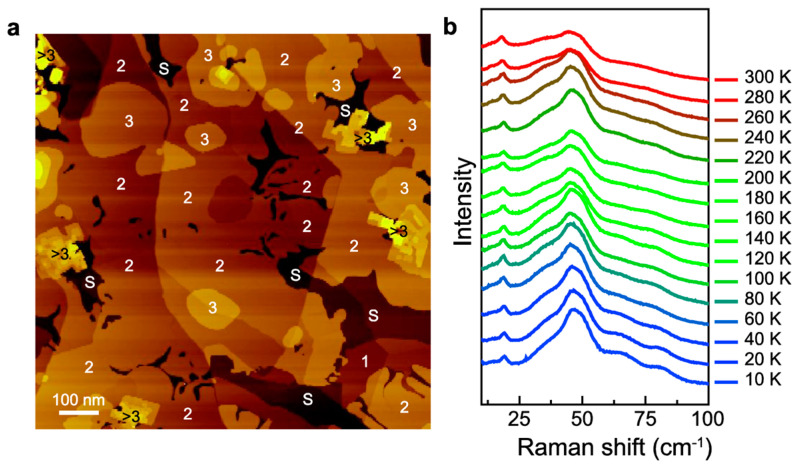
Structural and Raman characterization of ultrathin SnTe films: (**a**) STM topography of a SnTe film, with labeled thicknesses indicating substrate coverage distribution. (**b**) Raman spectra showing the TO mode at ~46.8 cm^−1^, persisting up to room temperature with slight peak shifts as temperature increases. Reproduced with permission from [119].

**Figure 18 materials-18-01530-f018:**
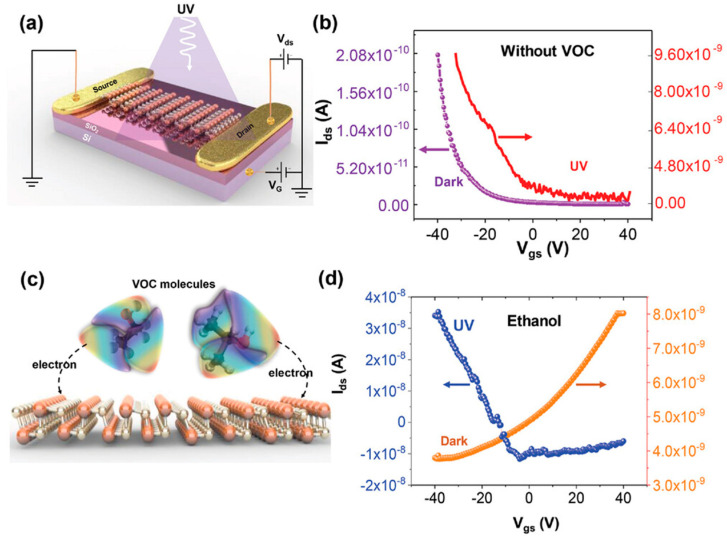
Mechanism of VOC sensing using a GeS FET-based sensor. (**a**) Schematic representation of a GeS FET under UV light irradiation. (**b**) Transfer curve (I_ds_ vs V_gs_) of the GeS FET under dark and UV light conditions, demonstrating p-type semiconducting behavior in the absence of VOCs. (**c**) Schematic illustration of charge transfer interactions between VOC molecules and the GeS surface. (**d**) Transfer curve of the GeS FET during ethanol exposure, showing a transition to n-type behavior in dark conditions, while maintaining p-type behavior under UV illumination. Reproduced with permission from [120].

**Figure 19 materials-18-01530-f019:**
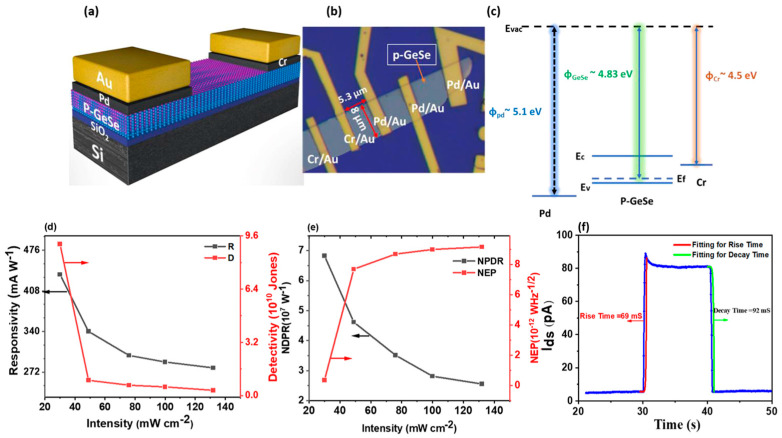
Self-powered near-infrared photodetector based on p-GeSe Schottky barrier diode. (**a**) Schematic illustration of a p-GeSe Schottky barrier diode photodetector. (**b**) Optical microscope image of the fabricated device. (**c**) Band diagram of Pd/p-GeSe/Cr. (**d**) Responsivity (R, mA W^−1^) and detectivity (D, Jones) as a function of light intensity. (**e**) Normalized photocurrent to dark current ratio (NPDR, W^−1^) and noise equivalent power (NEP, W Hz^−1^/^2^) under varying light intensities. (**f**) Time-dependent photoresponse, showing rise and decay times of 69 ms and 92 ms, respectively. Reproduced with permission from [125].

**Figure 20 materials-18-01530-f020:**
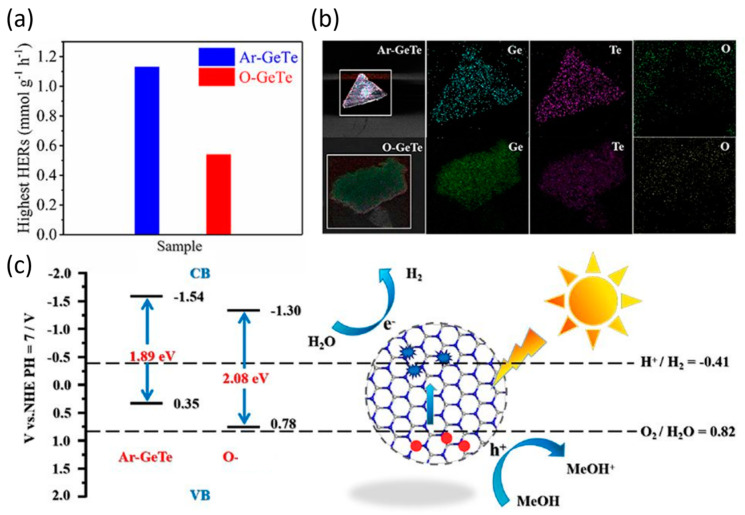
Photocatalytic hydrogen evolution performance of 2D GeTe nanosheets. (**a**) Hydrogen evolution rates (HERs) of Ar-GeTe and O-GeTe nanosheets, showing enhanced photocatalytic efficiency under argon-prepared conditions. (**b**) Energy dispersive spectroscopy (EDS) patterns of Ar-GeTe and O-GeTe nanosheets, confirming elemental composition and uniform distribution of Ge, Te, and O. (**c**) Energy band structure and photocatalytic process diagram, illustrating the mechanism of photocatalytic hydrogen production. Reproduced with permission from [60].

**Figure 21 materials-18-01530-f021:**
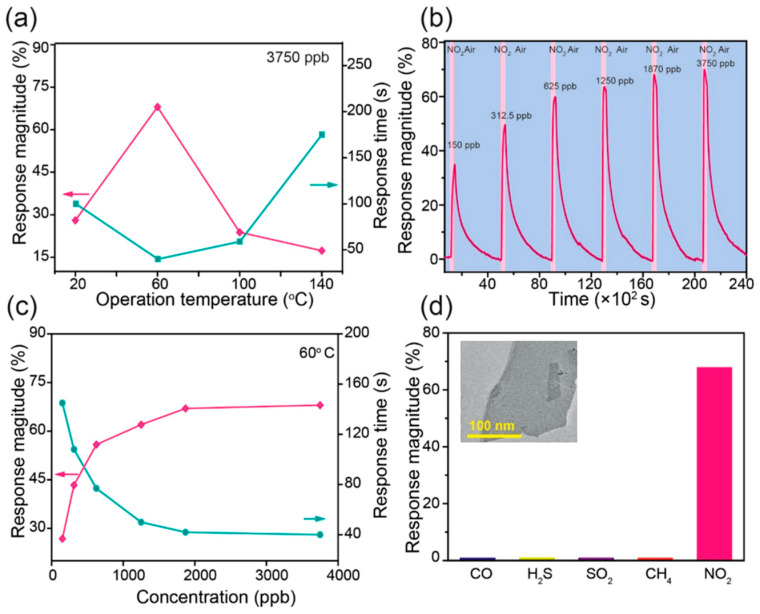
NO_2_ sensing performance of 2D SnS nanoflakes. (**a**) Response time and response magnitude of 2D SnS sensors under 3750 ppb NO_2_ exposure at varying operating temperatures. (**b**) Sensing performance of 2D SnS as a function of NO_2_ concentration at 60 °C. (**c**) Extracted response magnitude and response time demonstrating the sensitivity of the sensor. (**d**) Comparison of response magnitude of 2D SnS to CO (300 ppm), CH_4_ (1%), SO_2_ (100 ppm), H_2_S (20 ppm), and NO_2_ (3750 ppb) at 60 °C, highlighting its exceptional selectivity for NO_2_ detection. Reproduced with permission from [133].

**Figure 22 materials-18-01530-f022:**
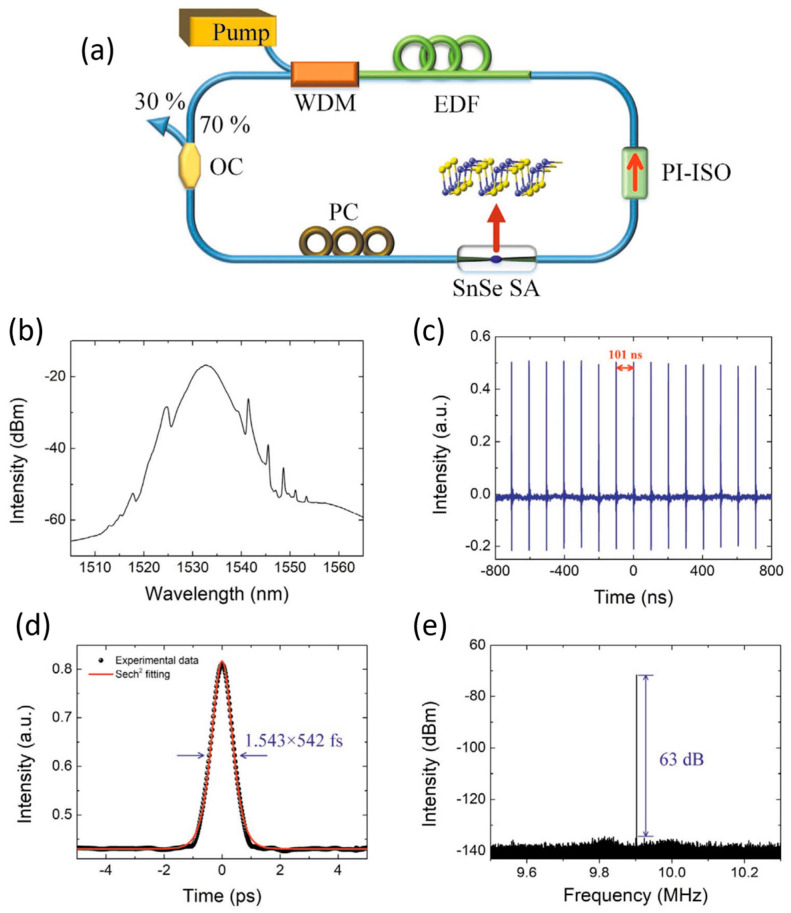
Mode-locked fiber laser based on a microfiber-integrated 2D SnSe saturable absorber. (**a**) Schematic illustration of an erbium-doped fiber laser (EDFL) mode-locked using a microfiber-based SnSe saturable absorber (SA). (**b**) Optical spectrum of the mode-locked pulses at 1532.76 nm, showing a 3 dB bandwidth of 5.08 nm. (**c**) Evolution of the temporal pulse trains, confirming a stable repetition rate of 9.9 MHz. (**d**) Autocorrelation trace of the pulses, revealing a pulse duration of 542 fs. (**e**) Radio-frequency (RF) spectrum, displaying a signal-to-noise ratio (SNR) of 63 dB, indicating stable mode-locked operation. Reproduced with permission from [140].

**Figure 23 materials-18-01530-f023:**
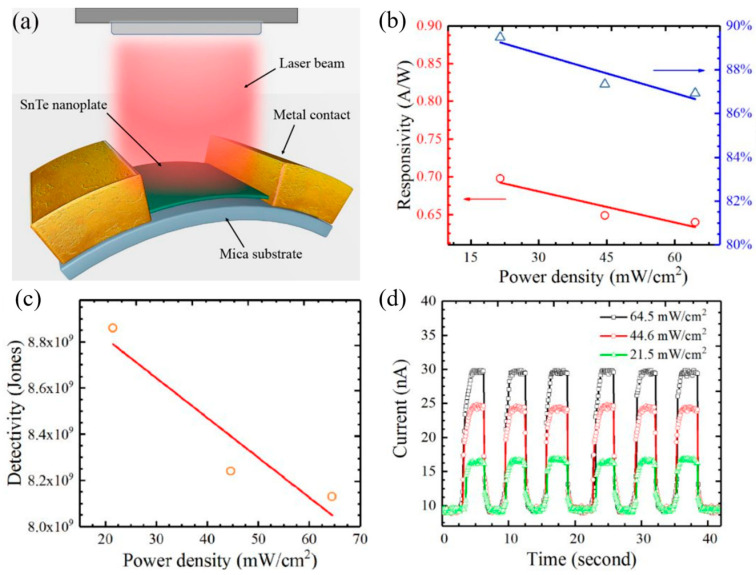
Flexible near-infrared photodetector based on ultrathin SnTe nanoplates. (**a**) Schematic diagram of a flexible SnTe NIR photodetector based on single nanoplates. (**b**) Responsivity (circle) and external quantum efficiency (EQE, triangle) of SnTe nanoplates under 980 nm laser illumination at a bias voltage of 0.5 V. (**c**) Specific detectivity (D) as a function of laser intensity, demonstrating high sensitivity. (**d**) Photoswitching behavior of the flexible SnTe nanoplate photodetector under 980 nm laser illumination, showing stable performance over multiple cycles. Reproduced with permission from [114].

**Table 1 materials-18-01530-t001:** Summary of the CVD growth conditions, material quality, and applications of 2D group IV monochalcogenides.

	Sintering Conditions	Material Quality	Application	References
GeS	GeS powder sublimation, diffusion-limited growth, reduced pressure, ~3–5 µm/min growth rate	Nanoflowers with nanosheet petals ~20–30 nm thick, ~100 µm lateral size	Photovoltaics	[36]
GeSe	GeI_2_ + Se precursors, 500 °C furnace, 3/7 sccm Ar/H_2_ carrier gas, atmospheric pressure	Ultrathin rectangular crystals, high-purity α-phase, 10–30 µm lateral size	Optoelectronics, anisotropic devices	[37]
GeTe	Ge + Te vaporized in one-zone furnace, ambient pressure, pre-annealed mica substrate	Single-crystalline nanosheets, ~8–9 nm thickness, up to ~30 µm lateral size	Ferroelectric devices	[39,40]
SnS	Two-step sulfurization of Sn thin film, ~200 °C, 110 mTorr, Ar flow ~100 sccm	Wafer-scale monolayer to few-layer films, 10–30 nm thickness, uniform morphology	Wearable piezoelectric devices	[41]
SnSe	Low-pressure vapor transport, Sn + Se precursors, controlled substrate temperature	Two- to four-layer sheets, single crystalline, up to ~23 µm lateral size	Photodetectors, ferroelectric devices	[43,44]
SnTe	SnI_4_/SnCl_2_ + Te precursors, 600–700 °C, H_2_/Ar carrier gas, substrate-dependent orientation	Single-crystalline nanosheets, 20–50 nm thick, several microns lateral size	Infrared photodetectors, topological crystalline insulators	[47]

**Table 2 materials-18-01530-t002:** Summary of the LPE and ME methods, material quality, and applications of 2D group IV monochalcogenides.

		Sintering Conditions	Material Quality	Application	References
GeS	LPE	Sonication in NMP, stable dispersions, oxidation-resistant basal planes	Few nm thick flakes, oxidation-resistant, basal plane stability	Photodetectors, photovoltaics	[53]
	ME	Scotch tape exfoliation, bulk crystal cleavage, CVT-grown crystals	Flakes 8–65 nm thick, high crystallinity, micron-scale lateral size	FETs, photovoltaics	[79]
GeSe	LPE	Sonication in isopropanol, two- to five-layer flakes, size-selective centrifugation	Few layers (≤5 layers), 15–180 nm lateral size, anisotropic structure	Optoelectronics, anisotropic transport	[55]
	ME	Micromechanical exfoliation, tape-assisted peeling, 14 nm thick flakes	14 nm thick flakes, low defect density, large lateral size	Photodetectors, anisotropic transport studies	[57]
GeTe	LPE	Sonication in ethanol, two to four layers, enriched by sequential centrifugation	Few-layer flakes (two to four layers), occasional monolayers, high surface area	Sensors, optoelectronics	[61]
	ME	Not feasible due to 3D bonding	N/A	N/A	N/A
SnS	LPE	Thermally assisted LPE, monolayer yield, few hundred nm lateral size	Monolayers to few layers, lateral size ~100–300 nm, stable morphology	Flexible electronics	[63,64]
	ME	Gold-assisted exfoliation, ~4.3 nm flakes, atomically smooth surfaces	4.3 nm thick, atomically smooth, lateral size ~10–30 µm	Flexible electronics, transistors	[65]
SnSe	LPE	Sonication in isopropanol, centrifugation for thickness control	2.5–8.9 nm flakes, 100–300 nm lateral size, sorted by centrifugation	Thermoelectric devices, IR detectors	[66,67]
	ME	Scotch tape exfoliation, 7 nm flakes, lateral sizes up to tens of microns	7–50 nm thick, tens of microns lateral size, anisotropic behavior	Thermoelectric applications, IR sensors	[72,73,74,75,76]
SnTe	LPE	Sonication in IPA, liquid dispersion of nanostructured SnTe	Nanoplatelets, few-layer thickness, enhanced thermoelectric response	Thermoelectric applications	[77]
	ME	Not feasible due to 3D bonding	N/A	N/A	N/A

**Table 5 materials-18-01530-t005:** Raman phonon modes of monochalcogenide phases stable at room temperature and ambient conditions (values in cm^−1^).

	GeS	GeSe	GeTe	SnS	SnSe	SnTe
A_g_	112 240 270	90 180 190	96	40 97.2 191 218	70 127 148	
B_3g_	213	153		50 163	105	
A_1_						123
E_TO_						61 139
E_g_			123			

**Table 6 materials-18-01530-t006:** Overview of applications, key performance metrics, and relevant references for 2D group IV monochalcogenides prepared by various synthesis techniques (ME, mechanical exfoliation; CVD, chemical vapor deposition; LPE, liquid-phase exfoliation; PLD, pulsed laser deposition; MBE, molecular beam epitaxy; PD, photodetector; SA, saturable absorber; NIR, near-infrared).

Material (Method)	Application	Key Performance	References
GeS (ME)—28 nm multilayer FET	Photodetector (broadband)	Responsivity: 206 A/W @633 nm (up to 655 A/W with V_g_) Detectivity: 2.35 × 10^13^ Jones; stable > 1 h operation	[11]
GeS (UV excitation, ME)	Gas sensor (VOC detection)	Selective response: distinct, VOC-specific current transients under UV illumination; enabled identification of VOC type (with ML analysis)	[120]
GeS (LPE)—Au NP hybrid	Flexible humidity sensor	Breath monitoring: real-time detection of human respiration; fast (<1 s) response and recovery to humidity changes (exhaled breath) reported	[121]
GeS (ME)—few layer	Ferroelectric photodiode	In-plane ferroelectricity: confirmed by hysteresis Bulk photovoltaic: switchable photocurrent under zero bias (shift current) upon polarization reversal	[122]
GeSe (polycrystalline)	Solar cell (thin-film)	Photoconversion: ~2–5% power conversion efficiency in early devices; band gap ~1.14 eV, high absorption coefficient (improvement needed via grain engineering)	[123]
GeSe (ME)—bulk crystal surface	Shift-current generator	THz emission: demonstrated ultrafast (>1 THz) surface photocurrent under 400/800 nm pulses (no bias) Use: THz source and photogalvanic detector	[124]
GeSe (ME)—few-layer p-GeSe	Self-powered photodetector	Broadband: 220–850 nm; *R* ≈ 0.28 A/W (850 nm, 0 V); D* ≈ 4 × 10^9^ Jones Response time: ~milliseconds; polarization-insensitive, stable cycling	[125]
GeTe (theory)—monolayer	Photocatalysis (water splitting)	DFT prediction: E_g_ ≈ 2.35 eV; high electron/hole mobility (~10^3^ cm^2^/V·s); visible-light absorption up to 10^5^ cm^−1^; suitable band edges for H_2_/O_2_ evolution	[59]
GeTe (LPE)—two- to four-layer GeTe	Photocatalysis (experiment)	H_2_ evolution: 1.13 mmol·g^−1^·h^−1^ (Ar-protected GeTe); 0.54 mmol·g^−1^·h^−1^ (oxide-coated GeTe); all thin GeTe samples have band positions to drive H_2_ generation	[60]
GeTe (exfoliated)—few layer	Ferroelectric memory (prospective)	Polarization: retains rhombohedral distortion in 2D; expected ferroelectric switching below ~300 K (extrapolated from bulk T_C_ ~700 K)	[144]
SnS (CBD)—nanoflakes on PET	Flexible UV–NIR photodetector	Spectral range: UV (380 nm) to NIR (850 nm) High sensitivity: e.g., ~3000% (UV) to ~446% (850 nm) at 3 V bias; fast response, good stability over bending	[147]
SnS (ME/CVD)—few layer	Polarization-sensitive PD	Anisotropy: pronounced polarization dependence in photocurrent (ratio > 2 for orthogonal polars at NIR) due to low symmetry; enables imaging of polarization state of incident light	[132]
SnS (solution exfoliation)	Saturable absorber (laser)	Nonlinear optics: mode-locking of fiber lasers at 1–2 µm achieved; few-layer SnS exhibits ultrafast third-order nonlinearity (absorbs intense pulses, enabling ~picosecond pulse generation)	[131]
SnS (ME)—quantum confined	Photoluminescent sensor	Exciton PL: strong, excitation-dependent photoluminescence in 2D SnS Chemical sensing: PL intensity shifts in presence of analytes (demonstrated for certain vapors); points to exciton-based chemical sensors	[133]
SnS (LPE/solvothermal)	Photothermal therapy	NIR absorption: SnS nanosheets efficiently convert 808 nm laser light to heat; achieved >50 °C temperature rise in solution Biomedical: ablated cancer cells in vitro with minimal toxicity	[135]
SnSe (ME/CVD)—few layer	Fast NIR photodetector	Broadband + polarized: 360–1550 nm range; R ~9.3 A/W @1064 nm; D* ~4.1 × 10^10^ Jones; response time on order of 10^−9^ s (GHz bandwidth) Polarization ratio: ~2.3 (at 1064 nm)	[141]
SnSe (LPE)—solution processed	Ultrafast photonics (SA)	Mid-IR mode locking: SnSe nanosheets used to generate femtosecond pulses in 2 µm fiber lasers; exhibited broadband saturable absorption from near- to mid-IR; no degradation observed over hours of laser operation (indicative of stability)	[140]
SnTe (CVD/MBE)—few layer	Flexible IR photodetector	Near-IR response: fast photo-switching under 808 nm illumination; integrated on PET with minimal performance loss on bending; on/off ratio > 10^3^ under modest bias (few volts)	[114]
SnTe (PLD epitaxy)—10 nm film	Broadband photodiode	Spectrum: visible to 10 µm IR (self-powered beyond 4 µm) Mechanism: SnTe/Ge heterojunction creates built-in field Usage: could serve as a room-temperature mid-IR detector (leveraging SnTe’s narrow gap)	[89]
SnTe (theory)—few layer	Spintronic device	Persistent spin texture: predicted in monolayer SnTe (inversion symmetry broken but mirror symmetry preserved)—leads to protected spin polarization states Ferroelectric control: switchable out-of-plane polarization to toggle spin texture (“on/off” spin field effect)	[144]
SnTe@MnO_2_ (wet chemistry)	Theranostic nanoplatform	Cancer therapy: SnTe core converts NIR to heat (photothermal therapy), MnO_2_ shell generates O_2_ in tumor (improving outcomes in hypoxic tumors); achieved significant tumor cell killing in vitro under 808 nm laser, with MRI contrast enhancement from MnO_2_	[145]
SnTe (thin film)—few layer	Thermoelectric power gen.	Quantum-size enhancement: 2D SnTe shows increased carrier mobility and power factor compared to bulk (due to modified band structure); suggests higher ZT at nanoscale; experimentally, SnTe nanofilms have achieved ZT > 1 at ~773 K (doped p-type)	[146]

## Data Availability

The original contributions presented in this study are included in the article. Further inquiries can be directed to the corresponding author.

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
