# Peer review of "Advances in 2D Group IV Monochalcogenides: Synthesis, Properties, and Applications"

_materials, 2025, doi:10.3390/ma18071530_

Round 1

Reviewer 1 Report

Comments and Suggestions for Authors

The manuscript titled “Advances in 2D Group-IV Monochalcogenides: Synthesis, Properties, and Applications” is a comprehensive review article focusing on IV-VI compounds. It primarily examines the structural and electronic properties of these materials, while also considering their principal applications.

The article is interesting, well-organized, and perfectly fits the journal's scope. I have only a few minor suggestions that could enhance its impact.

In the final part of the introduction, the authors correctly mention the challenges in synthesising these materials, noting that “defect engineering has expanded their applicability.” I fully agree with this point. However, it should be substantiated with some references. I recommend adding a paragraph to elaborate on this aspect, as defects in chalcogenides can create functionalities that broaden the applications of these materials (see, for example, ChemPlusChem 2021, 87, e202100562).

A brief initial list of the principal applications covered in section 6 would be useful for readers when discussing applications.

In the conclusion section, I recommend mentioning that IV-monochalcogenide materials can be highly performant even if some defects are present in their structure.

Reviewer 2 Report

Comments and Suggestions for Authors

The authors of the review paper titled “Advances in 2D group-IV monochalcogenides: Synthesis, properties and applications” evaluated the synthesis, structural properties, and applications of group-IV monochalcogenides. The paper is interesting, but authors need to consider the following before publishing in Materials.

  • In the abstract part, the term of black phosphorene should be replaced to phosphorene or black phosphorus nanosheet.
  • In the introduction part of the paper, before moving to the group-IV, authors need to highlight the 2D materials such as BP, Maxene, MoS2, etch. and their exfoliation method in one paragraph. Specifically, highlighting top-down and bottom-up method, liquid, mechanical, electrochemical, and bipolar exfoliation need to be highlight in this section. Authors can use: https://doi.org/10.1039/C8CS00254A, https://doi.org/10.1002/sstr.202000148.
  • Introduction part should focus more on why these group of 2D material is important for the research and industry and what are the pros and cons of using them. Then highlight why this review is important for the community. Please revise this part.
  • The format of the present study is more likely a summary of each study. Please provide a more scientific review paper by providing a schematic of exfoliation method for this group of 2D materials. Even the crystal structure of group-IV is much better than table 1 to improve the quality of paper.
  • In terms of different exfoliation methods, authors must provide a table in terms of previous methods, their efficiency, their final application, etc.
  • For the characterization part, what are the limited factors for analyzing these 2D materials? Please highlight it in the manuscript.
  • In the application part, authors should provide more figures from the literature to find out the performance of the material easier. Please revise it. A comparable table in terms of performance is also valuable.
  • In the last part of paper, authors need to provide a section with the title of perspective and future and provide their hypothesis and understanding about the future of group-IV monochalcogenides.

Reviewer 3 Report

Comments and Suggestions for Authors

In the section introducing synthesis methods, it is suggested to rearrange the content by first elaborating on the principles, advantages, and limitations of each synthesis method in detail, followed by specific examples of material synthesis. This approach allows readers to first gain a comprehensive understanding of the synthesis methods before examining their applications in different materials, making the logic clearer. For example, when introducing Chemical Vapor Deposition (CVD), the reaction mechanism, required conditions, and key principles such as how CVD controls film thickness and composition should be systematically explained. Then, comparative analyses can be conducted on its specific advantages and unique challenges in synthesizing different Group IV monochalcogenides (such as GeS, SnSe), such as the influence of source material diffusion on growth during GeS synthesis. This makes the explanation of the synthesis method more organized.

In the application section, for the current research status of each material's application, additional performance comparison data and analysis with existing materials in the same application field can be included. 

In the explanation of some key conclusions and data, more original references should be added to support them. For example, when mentioning certain materials' bandgap values or thermoelectric performance parameters, in addition to existing references, early reports or authoritative studies related to these topics can be cited to enhance the reliability of the data and the persuasiveness of the conclusions, allowing readers to trace the data sources and research background.

Some paragraphs have overly long sentences and complex structures, which may affect readability and comprehension. Long sentences can be appropriately split into shorter ones, simplifying sentence structures and emphasizing key information. During the writing process, greater attention should be paid to the logical connection between paragraphs. This can be achieved by adding transitional sentences or subheadings to enhance the coherence of the article.

Below are some relevant references for your consideration: Molecules 2023, 28(21), 7287; Molecules 2024, 29(20), 4974.

Round 2

Reviewer 2 Report

Comments and Suggestions for Authors

The paper is ready to publish in the present form.

Reviewer 3 Report

Comments and Suggestions for Authors

 Accept in present form.